# Isoprene and monoterpene simulations using the chemistry-climate model EMAC (v2.55) with interactive vegetation from LPJ-GUESS (v4.0)

Ryan Vella[1,2], Matthew Forrest[3], Jos Lelieveld[1,4], and Holger Tost[2]

[1]Atmospheric Chemistry Department, Max Planck Institute for Chemistry, Mainz, Germany
[2]Institute for Atmospheric Physics, Johannes Gutenberg University Mainz, Mainz, Germany
[3]Senckenberg Biodiversity and Climate Research Centre (SBiK-F), Frankfurt am Main, Germany
[4]Climate and Atmosphere Research Center, The Cyprus Institute, Nicosia, Cyprus

**Correspondence:** Ryan Vella (ryan.vella@mpic.de), Holger Tost (tosth@uni-mainz.de)

**Abstract.**

Earth system models (ESMs) integrate previously separate models of the ocean, atmosphere and vegetation in one comprehensive modelling system enabling the investigation of interactions between different components of the Earth system. Global isoprene and monoterpene emissions from terrestrial vegetation, which represent the most important source of volatile organic compounds (VOCs) in the Earth system, need to be included in global and regional chemical transport models given their major chemical impacts on the atmosphere. Due to the feedback of vegetation activity involving interactions with weather and climate, a coupled modelling system between vegetation and atmospheric chemistry is recommended to address the fate of biogenic volatile organic compounds (BVOCs). In this work, further development in linking LPJ-GUESS, a global dynamic vegetation model, to the atmospheric chemistry-enabled atmosphere-ocean general circulation model EMAC is presented. New parameterisations are included to calculate the foliar density and leaf area density (LAD) distribution from LPJ-GUESS information. The new vegetation parameters are combined with existing LPJ-GUESS output (i.e. LAI and cover fractions) and used in empirically-based BVOC modules in EMAC. Estimates of terrestrial BVOC emissions from EMAC's submodels ONEMIS and MEGAN are evaluated using: (1) prescribed climatological vegetation boundary conditions at the land-atmosphere interface; and (2) dynamic vegetation states calculated in LPJ-GUESS (replacing the "offline" vegetation inputs). LPJ-GUESS-driven global emission estimates for isoprene and monoterpenes from the submodel ONEMIS were 546 Tg yr$^{-1}$ and 102 Tg yr$^{-1}$, respectively. MEGAN determines 657 Tg and 55 Tg of isoprene and monoterpene emissions annually. The new vegetation-sensitive BVOC fluxes in EMAC are in good agreement with emissions from the semi-process-based module in LPJ-GUESS. The new coupled system is used to evaluate the temperature and vegetation sensitivity on BVOC fluxes in doubling $CO_2$ scenarios. This work provides evidence that the new coupled model yields suitable estimates for global BVOC emissions that are responsive to vegetation dynamics. It is concluded that the proposed model setup is useful for studying land-biosphere-atmosphere interactions in the Earth system.

# 1 Introduction

The land surface of the Earth is dominated by vegetation, with forests covering $\sim$42 million km$^2$ in tropical, temperate and boreal regions, making up $\sim$30% of the total land area (Bonan, 2008). The terrestrial biosphere is known to be a primary source of volatile organic compounds (VOCs) such as isoprene and various terpenes, accounting for around 90% of the total VOC emissions to the atmosphere (Guenther et al., 1995). The processes driving VOC emissions from plants are complex and not fully understood, however, BVOCs seem to play a role in protecting photosynthetic activity in plants from damage caused by reactive oxygen species, which are synthesised in leaves at high temperatures (Niinemets, 2010; Harrison et al., 2013; Lantz et al., 2019). BVOC emissions can also be triggered by other chemical, physical or biological stresses and processes; e.g. herbivory (Laothawornkitkul et al., 2008), signalling between organisms (Zuo et al., 2019), or also oxidative stress originating from the atmosphere (e.g. under elevated ozone concentrations, Sharkey et al., 2008). Plants emit an array of VOCs, but different plant species emit different compounds according to their evolutionary adaptation. For example, the emission of isoprene can be considered an evolutionary trait that benefits certain plant species in hot, dry environments (Taylor et al., 2018). Isoprene and monoterpenes are the most abundant species among the biogenic volatile organic compounds (BVOCs) (Kesselmeier and Staudt, 1999; Lathiere et al., 2006; Guenther et al., 2012), and their high reactivity exerts a significant influence on atmospheric composition (Atkinson, 2000). The atmospheric chemical lifetime of such BVOCs ranges from minutes to hours (Atkinson and Arey, 2003) as they quickly interact with tropospheric species including carbon monoxide, hydroxyl radicals, and ozone (Lelieveld et al., 1998; Granier et al., 2000; Poisson et al., 2000; Pfister et al., 2008), thus altering the atmosphere's oxidation capacity. BVOCs are also the primary precursor for secondary organic aerosols (SOA), which can exert a significant forcing on the radiative balance of Earth, both directly through scattering and absorption of solar radiation, and indirectly through changing cloud properties (Rap et al., 2013; Scott et al., 2014). SOA also contributes to change in the radiation balance by decreasing the solar near-surface direct radiation while at the same time increasing the diffusive radiation contribution (Wang et al., 2019).

The first BVOC models employed empirical relations describing isoprene emission rate dependencies on temperature and light (Guenther et al., 1991, 1993; Tingey et al., 1981), and monoterpene emission rate dependency on temperature (Evans et al., 1982; Lamb et al., 1987; Tingey et al., 1980, 1981). The formulations include a species or vegetation type-specific emission factor characterised from field or laboratory measurements (e.g. Lamb et al., 1985; Arey et al., 1991; Guenther et al., 1993) which is defined for arbitrarily chosen environmental conditions (usually 30°C and 1000 $\mu$mol photons m$^{-2}$ s$^{-1}$) (Grote and Niinemets, 2007). This approach has been extensively used to study different ecosystem types all over the world including desert (Geron et al., 2006), grassland (Bai et al., 2006), savanna (Guenther et al., 1999; Otter et al., 2003), Mediterranean (Cortinovis et al., 2005), tropical (Harley et al., 2004), temperate (Karl et al., 2003) and boreal forests (Westberg et al., 2000). Empirical algorithms are also presently used in well-established global BVOC models such as ONEMIS (Kerkweg et al., 2006) and MEGAN (Guenther et al., 2006). These modules are presently integrated into the modelling system considered in this study (EMAC).

The land-biosphere-atmosphere interface in models is fundamentally important to studying the fate of BVOCs in the atmosphere, yet, early models were designed to simulate single components of the Earth system in isolation, prescribing simple non-interacting boundary conditions at the interface. However, models have become increasingly coupled with dynamic multidirectional fluxes between the different models considered. This yielded a new category of models we now call Earth System Models (ESMs). ESMs are highly comprehensive tools ideal for modelling past and future climate change with biogeochemical feedbacks and also for studying biosphere-atmosphere interactions explicitly (Flato et al., 2014). To this end, several modelling studies have linked atmospheric chemistry-enabled models with dynamic vegetation models to investigate the impacts of changing vegetation cover on global atmospheric emissions, atmospheric chemistry, and future climate change (e.g. Levis et al., 2003; Sanderson et al., 2003; Naik et al., 2004; Lathiere et al., 2005; Arneth et al., 2007b).

Sporre et al. (2019) employed an ESM to investigate climate forcing caused by BVOC-aerosol feedbacks, where it was determined that increased BVOC emissions and subsequent SOA formation in future climate scenarios result in $-0.43$ W $m^{-2}$ stronger net cloud forcing and $-0.06$ W $m^{-2}$ forcing from direct scattering of sunlight. A new ESM that integrates the chemistry-climate model EMAC (Roeckner et al., 2006; Jöckel et al., 2005) with the dynamic global vegetation model (DGVM) LPJ-GUESS (Smith et al., 2001; Sitch et al., 2003; Smith et al., 2014) has been recently developed (Forrest et al., 2020). In a first study, the coupled model gave a good representation of worldwide potential natural vegetation distribution, despite some regional variations, especially at lower spatial resolutions. This study presents further model coupling of LPJ-GUESS within the EMAC modelling system with respect to vegetation-driven emissions. New vegetation parameters are computed from LPJ-GUESS variables and used as (online) input vegetation information for empirical-based BVOC modules (ONEMIS and MEGAN) in EMAC. The new vegetation-sensitive isoprene and monoterpene emission fluxes in EMAC are evaluated and compared against emissions from the semi-process-based module (Niinemets et al., 2002, 1999) in LPJ-GUESS. The new model configuration is then used to examine temperature and fertilisation effects in doubling $CO_2$ climate scenarios.

## 2 Methods

### 2.1 EMAC modelling system (v2.55)

The EMAC (ECHAM/MESSy Atmospheric Chemistry) model is a numerical chemistry and climate modelling system that includes submodels that describe tropospheric and middle atmosphere processes, as well as their interactions with oceans, land, and anthropogenic activities. It originally combined the ECHAM atmospheric general circulation model (GCM) (Roeckner et al., 2006) with the Modular Earth Submodel System (MESSy) (Jöckel et al., 2005) framework and philosophy where physical processes and most of the infrastructure has been divided into "*modules*", which can be further developed to improve existing process representations and new modules can be added to represent new or alternative process representations. EMAC has been further developed to include a broader representation of atmospheric chemistry by coupling different processes such as representations for aerosols, aerosol–radiation and aerosol-cloud interactions, e.g. Tost (2017). In this study, version 2.55 has been utilised, which is based on the well-documented version used in comprehensive model intercomparison studies, e.g. Jöckel et al. (2016).

**BVOC modules in EMAC**

Both ONEMIS and MEGAN are emission modules which are based on the Guenther algorithms (Guenther et al., 1993, 1995), where emissions are calculated as a function of ecosystem-specific emission factors, surface radiation, temperature, the foliar density and its vertical distribution. The schemes mostly differ in the evaluation of the canopy process for light and temperature sensitivity on emission yields. In ONEMIS, fluxes are a function of foliar density, plant-specific emission factors and an activity factor accounting for light and temperature sensitivity. Emissions are calculated within four distinguished layers of the canopy, expressed by the leaf area density (LAD) and leaf area index (LAI). For each layer, the extinction of photosynthetically active radiation (PAR) is calculated from the direct visible radiation and the zenith angle. The fractions of sunlit leaves and the total biomass are then used to calculate emissions from sunlit and shaded leaves within the canopy. On the other hand, fluxes in MEGAN are a function of the LAI, plant-specific emission factors, light, temperature and wind conditions within the canopy, leaf age, and soil moisture. In MEGAN, the parameterised canopy environment emission activity (PCEEA) algorithm is used rather than the alternative detailed canopy environment model that calculates light and temperature at each canopy depth. The PCEEA algorithm calculates the light sensitivity within the canopy as a function of the daily average above-canopy photosynthetic photon flux density (PPFD), the solar angle, and a non-dimensional factor describing the PPFD transmission through the canopy. Further technical details for canopy processes employed in ONEMIS can be found in Ganzeveld et al. (2002), while Guenther et al. (2006) provides details for MEGAN.

## 2.2 LPJ-GUESS DGVM (v4.0)

The following section is based on the standard copyright-free LPJ-GUESS model description template[1]. Lund-Potsdam-Jena General Ecosystem Simulator (LPJ-GUESS) (Smith et al., 2001; Sitch et al., 2003; Smith et al., 2014) is a DGVM featuring an individual-based model of vegetation dynamics. These dynamics are simulated as the emergent outcome of growth and competition for light, space and soil resources among woody plant individuals and a herbaceous understorey in each of a number (50 in this study) of replicate patches representing random samples of each simulated locality or grid cell. The simulated plants are classified into one of twelve plant functional types (PFTs) discriminated by growth form, phenology, photosynthetic pathway (C3 or C4), bioclimatic limits for establishment and survival and, for woody PFTs, allometry and life history strategy. LPJ-GUESS has already been implemented in global ESMs (e.g. Weiss et al., 2014; Alessandri et al., 2017), and more recently coupled with EMAC (Forrest et al., 2020). LPJ-GUESS coupled with EMAC currently provides information on potential natural vegetation rather than present-day vegetation, hence the current configuration cannot be validated yet. However, land use configurations are currently included in the coupled EMAC/LPJ-GUESS system allowing for a more realistic representation of the vegetation dynamics in upcoming studies.

**BVOC emission routine in LPJ-GUESS**

LPJ-GUESS includes a built-in BVOC emission module for the calculation of isoprene and monoterpene emission fluxes. The submodel combines the process-based leaf level emission model (Niinemets et al., 2002, 1999), which is also based on the Guenther algorithms, with the LPJ-GUESS vegetation model for isoprene (Arneth et al., 2007b) and monoterpene (Schurgers

---

[1]https://web.nateko.lu.se/lpj-guess/resources.html (last access: 14 January 2023)

et al., 2009) emissions. The algorithm computes BVOC production based on photosynthetic electron flux, emission factors, temperature, seasonality and also includes a $CO_2$ inhibition factor on leaf production of isoprene and monoterpenes relative to the $\sim 370$ ppmv $[CO_2]$ in the year 2000. Further technical details on the algorithm can be found in Hantson et al. (2017).

The algorithm also needs the daily temperature range (DTR) for the calculation of BVOC emission rates, which are typically derived from climatological datasets. In this study, the DTR (defined as the difference between the maximum and minimum daily temperature) is computed in EMAC and pass it on to LPJ-GUESS on a daily basis.

Semi-process-based BVOC emissions from the LPJ-GUESS module are only calculated within the LPJ-GUESS model part and are not integrated into or transferred to EMAC at the current stage. This also means that such emissions are only available as daily averages, in contrast to the emissions provided by ONEMIS and MEGAN in EMAC, which exhibit a diurnal cycle. Thus, having LPJ-GUESS-driven emissions from ONEMIS and MEGAN in EMAC provides more consistency including the direct link between weather, climate (change) and the impacts on vegetation and hence emissions. An adaptation of this LPJ-GUESS module to the shorter (a few minutes) time step of EMAC is rather complicated, especially, when the current scheme uses daily average light fluxes and a daily temperature range instead of individual snapshots of radiative fluxes and temperature. This would require a complete re-tuning of the emission scheme, with the only benefit of the higher temporal resolution of the emission fluxes (which cannot be utilised in LPJ-GUESS, but in EMAC only). Even though the scheme of Niinements et al. is semi-process-based, the processes are also highly parameterised, such that the advantages against the Guenther et al. algorithms are also small. In this work, BVOC emissions from the LPJ-GUESS routine are used for comparison only.

## 2.3  LPJ-GUESS-EMAC coupling for BVOC emission estimates

### 2.3.1  Overview of the coupling between EMAC and LPJ-GUESS

This study is part of a roadmap where the model integration between EMAC and LPJ-GUESS is gradually tightened in well-defined consecutive steps. To clarify the significance of this work, the roadmap from Forrest et al. (2020) is reproduced (Fig. 1), and the modeling development and evaluation efforts in this study are highlighted. This study focus on BVOC model processes in EMAC based on interactive vegetation from LPJ-GUESS. It extends on the model coupling between EMAC and LPJ-GUESS in Forrest et al. (2020) by employing new parameterisations to calculate the foliar density and leaf area density distribution from vegetation states in LPJ-GUESS. The new parameters from LPJ-GUESS are combined with existing ones (e.g. LAI and cover fractions) and are used to run empirical-based BVOC modules in EMAC (i.e. ONEMIS & MEGAN). The semi-process-based BVOC emissions from LPJ-GUESS are not integrated into EMAC and are only evaluated against the new vegetation-sensitive empirical-based emissions fluxes in EMAC.

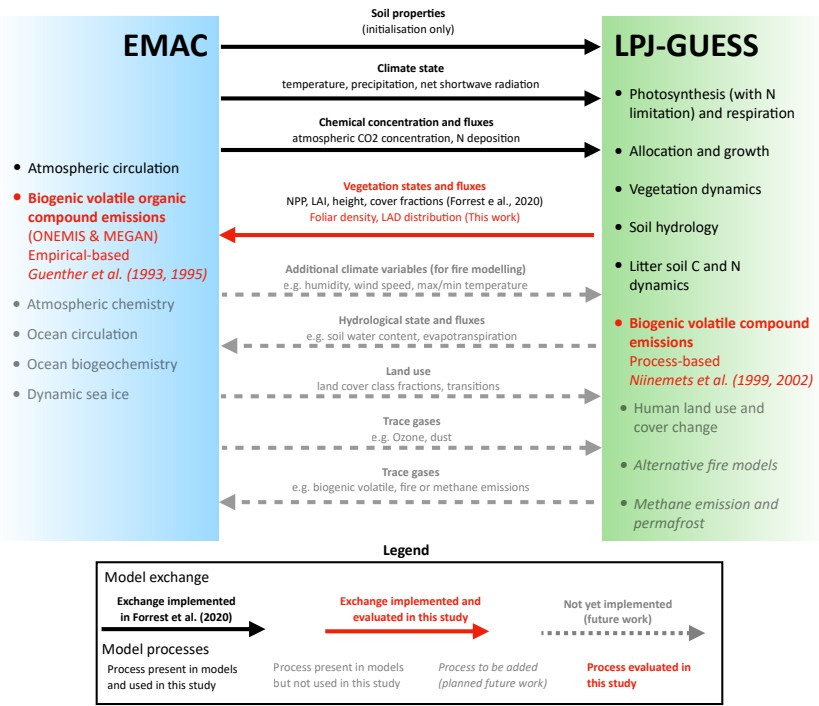

**Figure 1.** The main processes and exchanges in the coupled model framework adopted from Forrest et al. (2020). Model exchanges in normal black text were included in Forrest et al. (2020) and used in the simulations presented here. Exchanges in red are implemented in this work, while exchanges with grey text and grey dotted arrows are not currently included in the framework but are planned in future work. Model processes in normal black text were included in Forrest et al. (2020). Processes in normal grey text were included in Forrest et al. (2020) but not used here, while processes in italic grey are not currently included in the framework. Processes in red on the EMAC side (empirical-based emissions with interactive vegetation information) are implemented and evaluated in this work while BVOC emissions in LPJ-GUESS (semi-process-based) were already implemented and are only used in this study for comparison with EMAC emissions.

### 2.3.2 Simulation setup

In this work, the coupling strategy employed in Forrest et al. (2020) is employed, where modifications are done in LPJ-GUESS such that it provides its functionality (i.e. vegetation information) via a new submodel in the MESSy framework, yet keeping the LPJ-GUESS source code intact with minimal modifications. At a regular interval (currently 24 hours at 12 UTC) EMAC provides LPJ-GUESS the daily-mean 2m temperature, daily-mean net downwards shortwave radiation, and the total daily precipitation. Daily $CO_2$ concentrations and nitrogen deposition are also transferred from EMAC to LPJ-GUESS.

As a result, the LPJ-GUESS land surface conditions are entirely determined by the EMAC atmospheric state and chemical fluxes. In Forrest et al. (2020), only one-way coupling was performed (see Fig. 1), which means that LPJ-GUESS calculations are entirely based on EMAC's information, however, the land surface vegetation condition calculated in LPJ-GUESS has no effect on the atmospheric state in EMAC. In this study, existing LPJ-GUESS output variables (i.e. LAI and PFT fractional

coverage) are utilised to calculate isoprene and monoterpene emission rates in EMAC, allowing for the first time dynamic
vegetation information from LPJ-GUESS to be passed back to EMAC and used for BVOC fluxes calculations in ONEMIS
and MEGAN. Note, that the hydrological cycle uses the ECHAM5 native soil moisture and the hydrological information from
LPJ-GUESS does not feed back on the meteorology. Similarly, plant albedo and roughness height are not allowed to influence
the meteorology either, but climatological values are used to drive the weather and climate conditions of EMAC in the applied
model configuration, even though these links are also implemented and pending throughout evaluation.

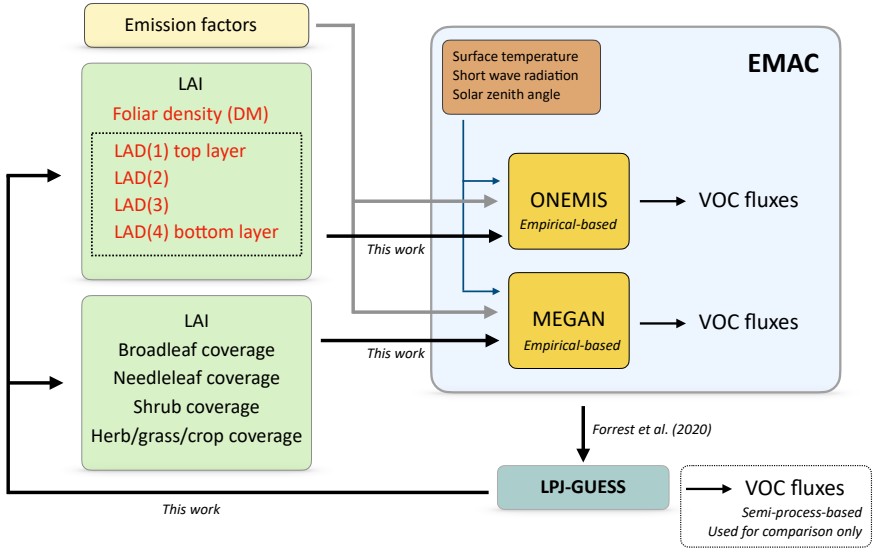

**Figure 2.** Model setup for BVOC emission estimates in EMAC. The vegetation variables needed in ONEMIS and MEGAN are now provided
by LPJ-GUESS, replacing offline climatological datasets. Vegetation variables that were already available at the interface between EMAC and
LPJ-GUESS are in normal black text while the new vegetation information derived in this work (i.e. Foliar density and the LAD distribution)
are in red text. Note that in EMAC only empirical-based emissions are considered. VOC fluxes from the semi-process-based module in
LPJ-GUESS are only used for comparison.

Fig.2 illustrates the model configuration for computing isoprene and monoterpene emissions fluxes in EMAC using the
submodels ONEMIS and MEGAN. Both models require emission factors for the various PFTs, the solar zenith angle, surface
radiation, and surface temperature. Additionally, ONEMIS requires the following *vegetation* variables: leaf area index (LAI),
foliar density, and leaf area density (LAD) canopy profile. In contrast, MEGAN requires the LAI and fractional coverage of
broadleaf, needleleaf, grass and shrub ecosystem types. In the original setup, the vegetation input variables are prescribed
from offline climatological datasets, whereas, in the new configuration, the climatological vegetation variables are replaced
with ones calculated online in LPJ-GUESS. This implies that the new setup feeds dynamic vegetation states to the BVOC
modules that are directly computed in LPJ-GUESS on a daily time scale and driven by atmospheric states and chemical fluxes
in EMAC, allowing for estimates of isoprene and monoterpene emissions with dynamic vegetation. The coupling is performed
from the EMAC side by implementing new calculations using LPJ-GUESS' information to derive all vegetation variables

needed in ONEMIS and MEGAN for the computation of BVOC emission fluxes. Details on the vegetation variables required in ONEMIS and MEGAN, as well as the new parametrisations used for their calculation, are described in the next section.

### 2.3.3 Vegetation variables for the BVOC modules

**Leaf area index:** Measurements of the amount of leaves in the canopy are required for ecosystem studies such as this one. This metric is often defined as the leaf area index (LAI), which is the one-sided leaf area in the canopy per unit surface area of 180 the ground ($m^2$ $m^{-2}$) (Jordan, 1969). In DGVMs, including LPJ-GUESS, this is a standard output variable.

**Foliar density:** The foliar density $D$ (g dry matter $m^{-2}$), sometimes referred to as dry matter (DM), can be derived directly from the LAI as follows (Guenther et al., 1995):

$$D = \text{LAI} \cdot S_{LW} \tag{1}$$

where $S_{LW}$ is an average specific leaf weight (g $m^{-2}$) and is given for each ecosystem (or PFT) based on Box (1981).

**Leaf area density distribution:** The leaf area density (LAD) is a metric describing the leaf area in a cubic volume within the canopy ($m^2$ $m^{-3}$). The original ONEMIS configuration employs an *expert-driven* offline dataset from a dry deposition inferential model (DDIM) (Weiss and Norman, 1985) to characterise the LAD distribution for three types of vegetation: (i) agricultural crops, (ii) deciduous forests, and (iii) coniferous forests. The twelve PFTs (used in the applied LPJ-GUESS setup) were classified into these three groups, with grass PFTs included in the "agricultural crop" category. The LAD distribution for 190 each of the vegetation types is divided into four equal layers; LAD 1 (top layer), LAD 2, LAD 3 and, LAD 4 (bottom layer).

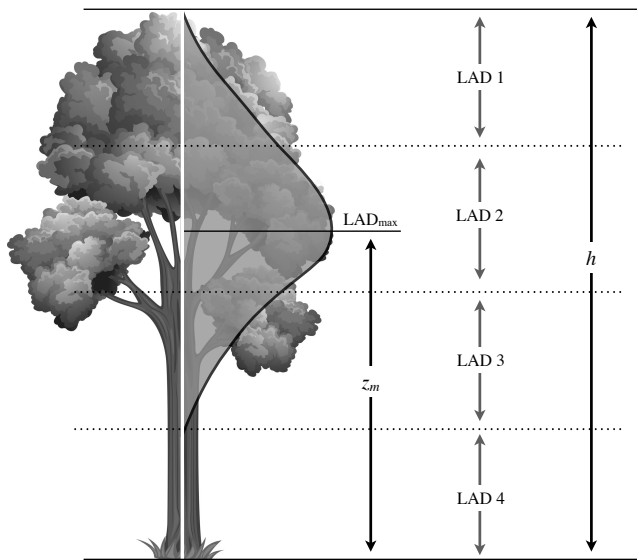

**Figure 3.** Graphical representation of the canopy LAD distribution. $\text{LAD}_{max}$ is the maximum LAD within the canopy, $z_m$ is the height from the ground at which $\text{LAD}_{max}$ occurs, and $h$ is the total canopy height. LAD 1, LAD 2, LAD 3, and LAD 4 are four equal canopy layers.

These values are then used in ONEMIS for calculating the photosynthetic active radiation (PAR) within the canopy and subsequent BVOC emission estimates. For the LPJ-GUESS output, a parametrisation derived by Lalic et al. (2013), is employed to compute similar LADs at four canopy levels in the interface submodel between EMAC and LPJ-GUESS. The empirical relation describes the LAD at a height $z$ (in m) as a function of the maximum LAD ($LAD_{max}$), the canopy height $h$, and the height $z_m$ corresponding to $LAD_{max}$ (see Fig. 3).

$$\text{LAD}(z) = \text{LAD}_{max} \left( \frac{h - z_m}{h - z} \right)^n \exp\left[ n \left( 1 - \frac{h - z_m}{h - z} \right) \right] \qquad \begin{array}{ll} n = 6 & \text{for} \quad 0 \le z < z_m \\[4pt] n = \frac{1}{2} & \text{for} \quad z_m \le z \le h \end{array} \tag{2}$$

Firstly, the ratio between the canopy height ($h$) and the height corresponding to $LAD_{max}$ (i.e. $h/z_m$) for each vegetation class using LAD canopy profiles from the DDIM is determined. The dataset has 21 layers, and the layer where $LAD_{max}$ occurs ($z_m$) is utilised to compute $h/z_m$ as follows:

(i) Agricultural crops : $z_m/h = 12/21 \approx 0.57$

(ii) Deciduous forests : $z_m/h = 15/21 \approx 0.71$

(iii) Coniferous forests : $z_m/h = 17/21 \approx 0.81$

$LAD_{max}$ for each PFT is calculated from the corresponding LAI, and PFT height $h$ information from LPJ-GUESS, as well as the ratio $h/z_m$, given the relation:

$$\text{LAI} = \int_0^h \text{LAD} = \int_0^h \text{LAD}_{max} \left( \frac{h - z_m}{h - z} \right)^n \exp\left[ n \left( 1 - \frac{h - z_m}{h - z} \right) \right] dz \tag{3}$$

After a numerical value for $LAD_{max}$ for each PFT is computed, the LAD at four canopy layers is calculated via Eq. 2 by integrating over four equal layers within the total canopy height $h_{tot}$. In all setups used in the study, the total canopy height $h_{tot}$ is assumed to have a value of 25 m. This results in:

$$\text{LAD}(1) = \int_{0.75h_{tot}}^{h_{tot}} \text{LAD}_{max} \left( \frac{h - z_m}{h - z} \right)^n \exp \left[ n \left( 1 - \frac{h - z_m}{h - z} \right) \right] dz,$$

$$\text{LAD}(2) = \int_{0.5h_{tot}}^{0.75h_{tot}} \text{LAD}_{max} \left( \frac{h - z_m}{h - z} \right)^n \exp \left[ n \left( 1 - \frac{h - z_m}{h - z} \right) \right] dz,$$

(4)

$$\text{LAD}(3) = \int_{0.25h_{tot}}^{0.5h_{tot}} \text{LAD}_{max} \left( \frac{h - z_m}{h - z} \right)^n \exp \left[ n \left( 1 - \frac{h - z_m}{h - z} \right) \right] dz,$$

$$\text{LAD}(4) = \int_{0}^{0.25h_{tot}} \text{LAD}_{max} \left( \frac{h - z_m}{h - z} \right)^n \exp \left[ n \left( 1 - \frac{h - z_m}{h - z} \right) \right] dz.$$

where $z_m$ is a fraction of $h$ based on the PFT's vegetation class i, ii, or iii, and $h$ is the PFT height. Fig. 4 compares the LAD distribution from DDIM point data, used in the previous setup, with the new parametrisation described in Eq. 2.

**Vegetation class coverage:** The vegetation coverage refers to the fraction of land area occupied by certain PFTs in one grid cell. This variable is used in MEGAN to adjust emission rates from different vegetation classes. This variable is already calculated in LPJ-GUESS for each of the twelve PFTs.

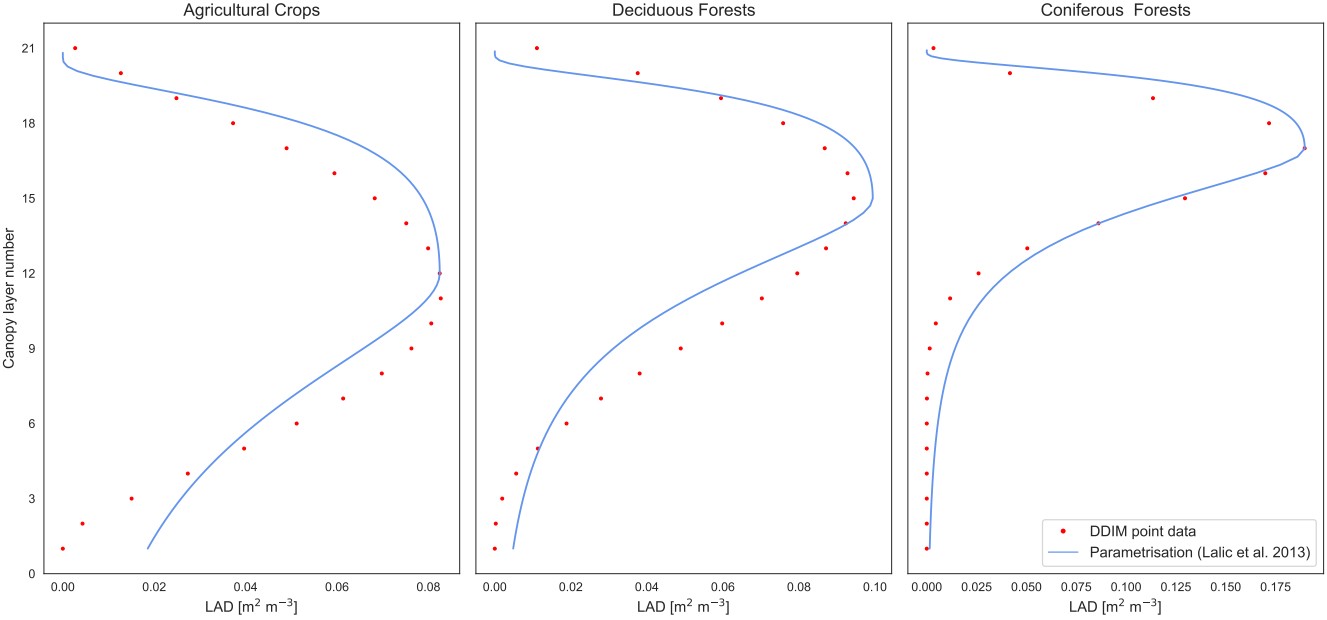

**Figure 4.** LAD distribution for a 21-layer canopy using DDIM point data versus the continuous distribution from the employed parametrisation for the three vegetation classes.

## 2.4 Setup for double CO$_2$ scenarios

The submodel RAD in EMAC (Dietmüller et al., 2016) simulates the radiative transfer in the atmosphere accounting for the effects of the shortwave and longwave radiation fluxes from radiatively active trace gases. CO$_2$ has the largest radiative influence in the longwave range of the spectrum, resulting in radiative forcings leading to stratospheric cooling and tropospheric warming. The CO$_2$ value prescribed in RAD mainly dictates surface temperatures resulting from the greenhouse effect, while CO$_2$ in the vegetation scheme (i.e. in LPJ-GUESS) determines the carbon available for photosynthesis and hence accounts for CO$_2$-fertilisation effects.

Climatological monthly average sea surface temperature (SST) and sea ice content (SIC) from the AMIP database from 1987 to 2006 are used for Ref and Bio×2, with a prescribed CO$_2$ of 348 ppmv in the radiation scheme. However, with 696 ppmv [CO$_2$] in the radiation scheme (i.e. in Atm×2 and Both×2), oceanic boundary conditions are prescribed using external data corresponding to SST and SIC at 696 ppmv to preserve radiative equilibrium. This data is acquired from a coupled atmosphere-ocean general circulation model (increased/decreased SSTs/SICs) performed under identical climate circumstances (696 ppmv [CO$_2$]) (same approach as in Rybka and Tost (2014)).

## 3 Results and discussion

### 3.1 Vegetation characteristics as input to the emission routines

In this section, the LPJ-GUESS state variables needed as input for the BVOC routines are discussed. For that purpose, the LPJ-GUESS output is compared with the corresponding offline climatological datasets used for BVOC emission estimates in ONEMIS and MEGAN in the original model configuration i.e. with prescribed vegetation boundary conditions. In such comparisons it has to be kept in mind that the neglect of land use in LPJ-GUESS means that currently only the natural biosphere is represented; however, climatological values for LAI also do not fully account for managed land. Nevertheless, the impact of crops is likely to modify the vegetation representation and emission results to a certain degree. The simulation results presented are from a 10-year temporal average with a 500-year offline spin-up phase at a horizontal resolution of T63 (approximately 1.9°× 1.9°at the Equator). The simulations are climatological, meaning that the same boundary conditions are used for each year of the simulation.

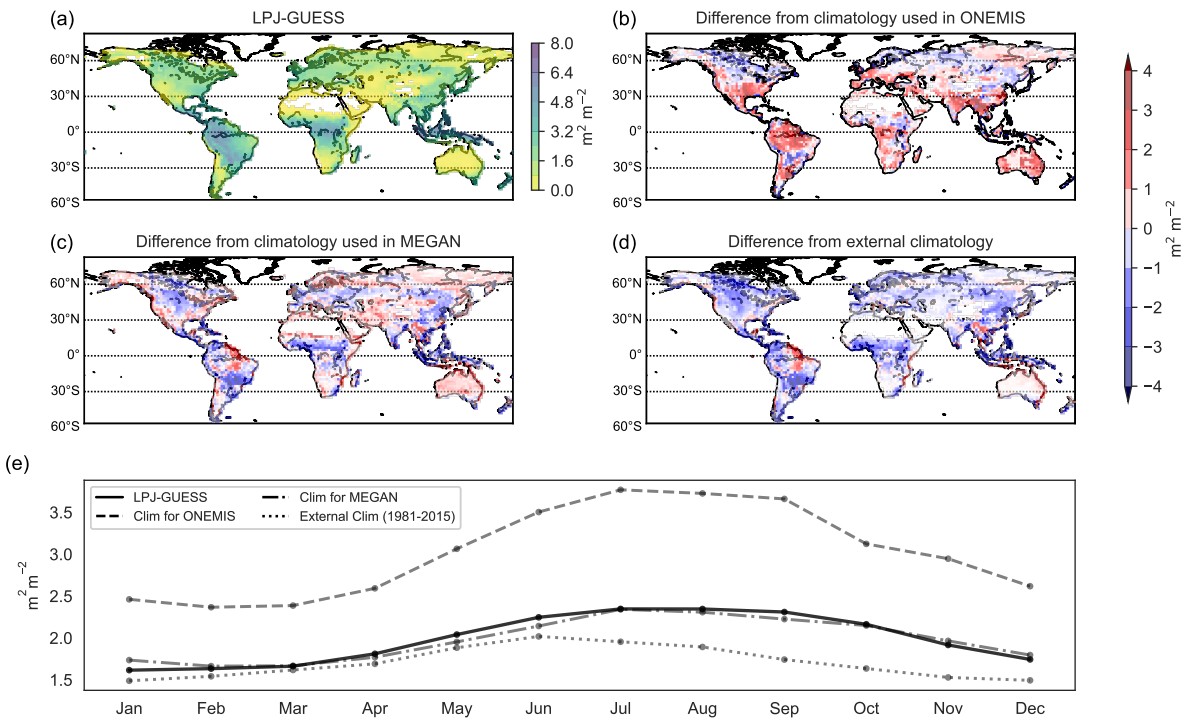

**Figure 5.** Geographic distribution of the LAI from LPJ-GUESS (panel a), as well as difference plots (LAI from LPJ-GUESS minus climatological LAI) from climatology used in the standard configuration in ONEMIS and MEGAN (panels b and c). An additional LAI climatology from 1981 to 2015 is also included (panel d). Panel e compares the global monthly averages of all datasets.

The spatial distribution of the LAI from LPJ-GUESS - shown in Fig. 5 - indicates elevated LAI of more than 6 m$^2$ m$^{-2}$
in the tropical rain forests of the Amazon, Congo and South East Asia. The LPJ-GUESS output is also compared with (1)
the LAI prescribed for ONEMIS; (2) the LAI prescribed for MEGAN; and (3) an external climatology dataset of the global
monthly mean LAI averaged over the period from August 1981 to August 2015 (Mao and Yan, 2019). This product uses
remotely sensed satellite data from the Moderate Resolution Imaging Spectroradiometer (MODIS) and the Advanced Very
High Resolution Radiometer (AVHRR) instruments. Note, that each emission scheme utilises its own climatological LAI and
an exchange of this dataset results in substantially modified emission fluxes. The difference plots indicate that the climatological
LAI used in ONEMIS is generally higher across the globe while the LAI climatology used in MEGAN is generally lower,
especially over the tropics. The external climatological dataset prescribes lower LAI compared to the LAI from LPJ-GUESS.
Panel (e) in Fig. 5 shows the global monthly mean LAI from all datasets. LPJ-GUESS exhibits a difference of 0.73 m$^2$ m$^{-2}$
between the lowest and highest month, while the variability is 1.40 m$^2$ m$^{-2}$ for the ONEMIS dataset, 0.68 m$^2$ m$^{-2}$ for the
MEGAN dataset, and 0.53 m$^2$ m$^{-2}$ for the external LAI dataset. This means that the LAI variability from LPJ-GUESS is
comparable to the LAI climatology used in MEGAN as well as the external LAI dataset, however, it significantly differs from
the variability in the climatological LAI used in ONEMIS. The foliar density is not presented here but, by definition, it is a

function of the LAI (Eq. 1), and hence shows a similar spatial and temporal distribution. Even though LPJ-GUESS provides potential natural vegetation rather than all present vegetation, a good agreement with climatological LAI is found, particularly the LAI climatology used in MEGAN. Consequently, it is concluded that the current LPJ-GUESS vegetation representation is adequate for estimating BVOC fluxes in ONEMIS and MEGAN, but that incorporating land use into the LPJ-GUESS-EMAC configuration will improve current (and future) representations of vegetation, as well as the resulting BVOC estimates.

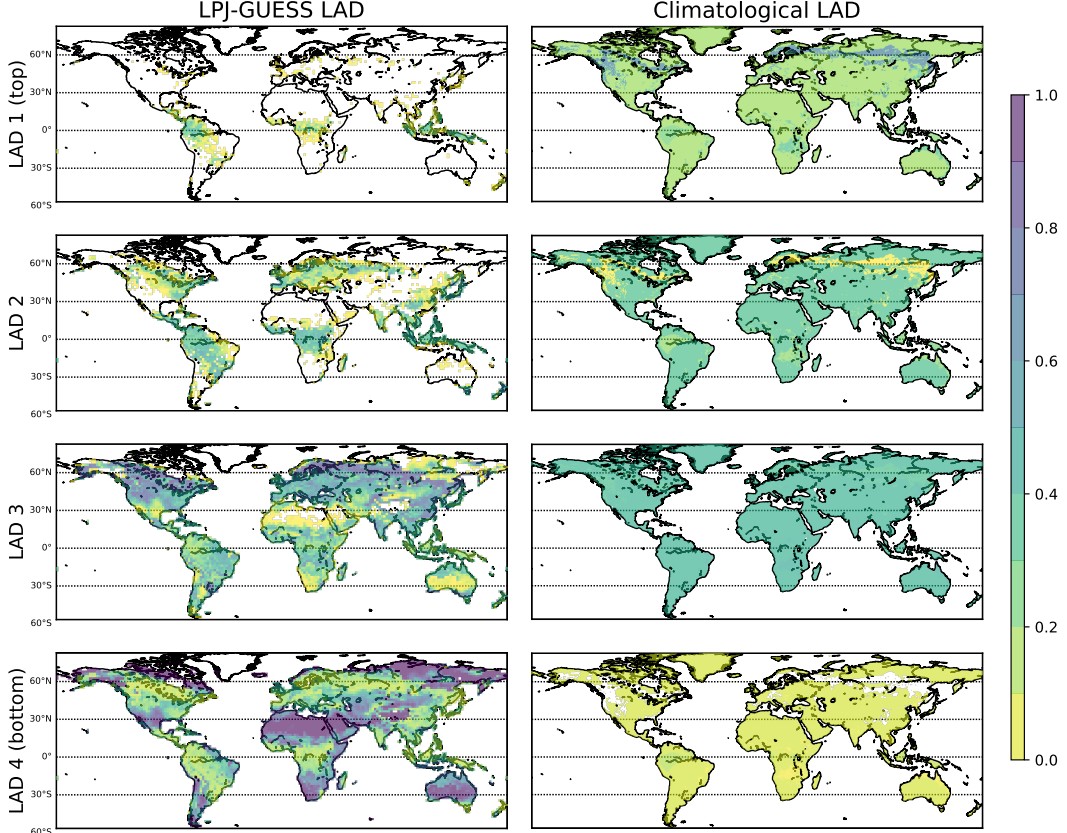

**Figure 6.** LAD fraction at four canopy layers from LPJ-GUESS (left panels) and the DDIM model (right panels). While the climatological LAD is oversimplified and homogeneous, LPJ-GUESS provides a higher-resolved LAD distribution at the four canopy layers.

Fig. 6 displays the LAD distribution derived from LPJ-GUESS and the climatology. The climatological LAD (derived from the DDIM model) has been used in the original setup in ONEMIS to calculate BVOC emission estimates whereas the LAD from LPJ-GUESS is derived from the parametrisation discussed in section 2.3.3. In LPJ-GUESS, a total canopy height of 25 m was assumed and thus the four LAD layers are classified as follows: bottom canopy layer (LAD 4) represents the LAD between 0-6.25 m, LAD 3 from 6.25-12.50 m, LAD 2 from 12.50-18.75 m, and the top canopy layer (LAD 1) from 18.75-25 m. Values of the four LAD layers (i.e. total canopy) add up to one. The four panels on the right in Fig. 6 show the geographic distribution of LAD from climatology (DDIM model). Even though, the LAD canopy distribution from climatology data is over-simplified

it indicates higher densities in leaf area in the uppermost layers of the canopy over the forested regions found in the tropics and boreal forests in the northern hemisphere. The panels on the left show the LAD geographic distribution from LPJ-GUESS at T63. With this approach, a better-resolved geographic distribution for the LAD at the different canopy layers is simulated. Increased LAD in the bottom layer (i.e. all vegetation below 6.25 m) highlights grasslands and desert areas. The layers above show how the LAD changes in the upper sections of the canopy, with plant biomass higher than 20 meters mostly found in

tropical forest areas. While the climatological LAD remains constant, the LAD from LPJ-GUESS changes from month to month. On average, climatological LAD 1 = 0.21, LAD 2 = 0.32, LAD 3 = 0.40, and LAD 4 = 0.08. LAD 1 from LPJ-GUESS stays constant at around 0.02, LAD 2 varies between 0.12 (January) and 0.18 (July). LAD 3 and LAD 4 vary a bit but remain close to 0.44 and 0.50, respectively, with no apparent seasonal trends.

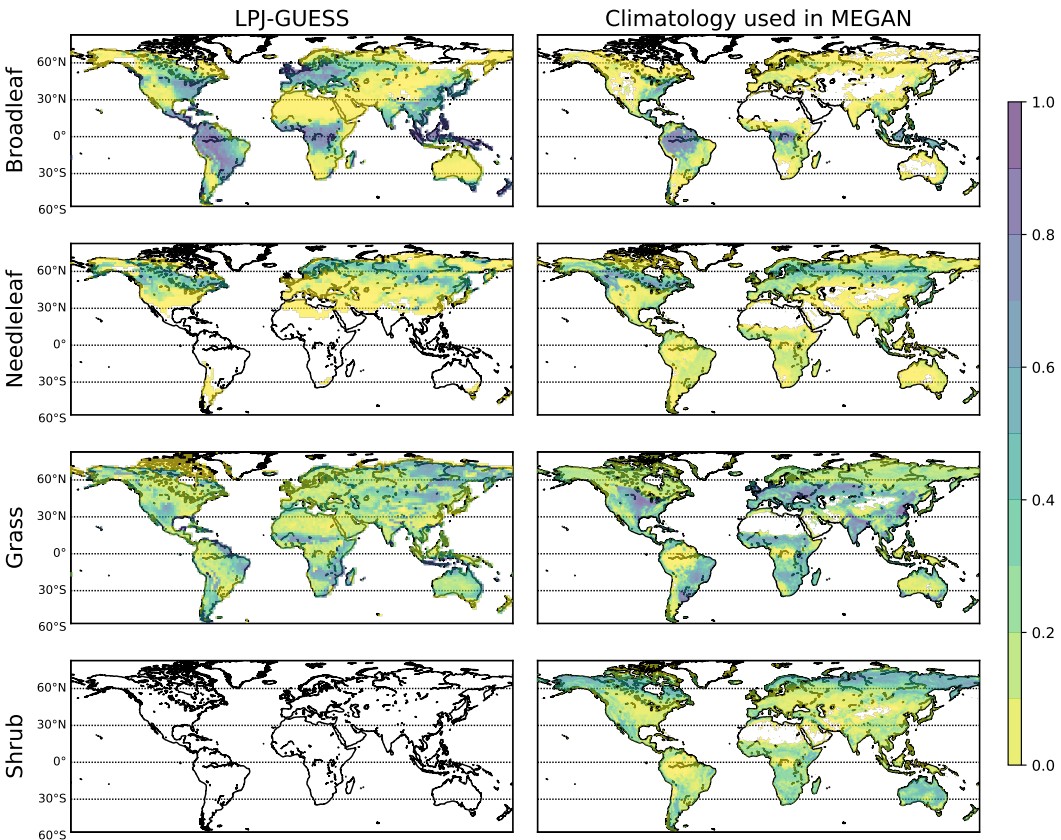

**Figure 7.** Fractional coverage for different vegetation classes used as inputs in MEGAN. The fractional coverage from LPJ-GUESS is displayed on the left panels, while the climatological fractional coverage is displayed on the right panels. Shrub PFTs are currently missing in the applied LPJ-GUESS global PFT set.

Fig. 7 shows the fractional coverage required as input for MEGAN. The twelve PFTs from LPJ-GUESS are classified

into the four vegetation classes: broadleaf trees, needleleaf trees, grass and shrub, and are compared to the corresponding

climatology vegetation fraction. The climatological fractional coverage remains constant throughout the whole simulation, while the fractional coverage from LPJ-GUESS updates from year to year. The global averages of climatological fractional coverage for broadleaf trees, needleleaf trees, grass, and shrubs are 0.13, 0.11, 0.23, and 0.14, respectively. LPJ-GUESS provides fractional coverage of 0.35, 0.11, 0.23, and 0 for broadleaf trees, needleleaf trees, grass, and shrubs, respectively. Note, that shrubs are not included in the currently applied LPJ-GUESS global PFT set, consequently, they are not considered in the applied simulation setup. Studies by (Forrest et al., 2015) did not use explicit shrub PFTs as well, and only in more recent, they are explicitly included (e.g. Allen et al., 2020). Even though this leads to less competition among some PFTs in certain regions, this is a limitation of the current study. However, including the new shrub PFTs is planned for future studies. Also, even though temperate needle-leaved trees exist in LPJ-GUESS, they are very uncompetitive and thus not well-represented in mixed forests.

## 3.2 Global isoprene and monoterpene emissions

### Isoprene emissions

Fig. 8 presents global isoprene emissions from ONEMIS and MEGAN with dynamic vegetation states from LPJ-GUESS as well as offline climatological vegetation inputs, at spatial resolution T63. The simulations have been conducted with fixed prescribed $CO_2$ of 348 ppmv (in both the radiation and vegetation schemes). Apart from the exchange of the *vegetation*, input parameters for the BVOC modules, the simulation setup, model code and parameter settings are identical in all simulations. All calculations are climatological, i.e., with identical boundary conditions for all years, and the results presented here are from 10 ensemble years.

Panels a and b in Fig. 8 show the geographic distribution of isoprene emission rates (in mg m$^{-2}$ day$^{-1}$) using LPJ-GUESS vegetation inputs. Strong isoprene emission fluxes can be seen over South America, Central Africa, Southeast Asia, and North Australia mostly corresponding to high vegetation densities in tropical rainforests. MEGAN emissions are significantly higher with up to 90 mg m$^{-2}$ day$^{-1}$ over the Amazon forest. Such differences in emissions between ONEMIS and MEGAN result from different canopy processes employed by the BVOC modules (see Section 2.1). Emission values shown in panels a and b are from the same simulation, meaning that input parameters (e.g. temperature, LAI, etc.) in ONEMIS and MEGAN are identical. Panels c and d compare our new emissions using LPJ-GUESS vegetation with emissions using climatological inputs. For both ONEMIS and MEGAN, emissions with dynamic vegetation are lower over tropical areas but higher over Australia compared to emissions using climatology. With climatological vegetation inputs, ONEMIS exhibits low but significant emissions over deserted regions, particularly the Saharan desert in North Africa, resulting from the poor representation of the LAD distribution in the original model setup. This is not the case when using dynamic vegetation inputs.

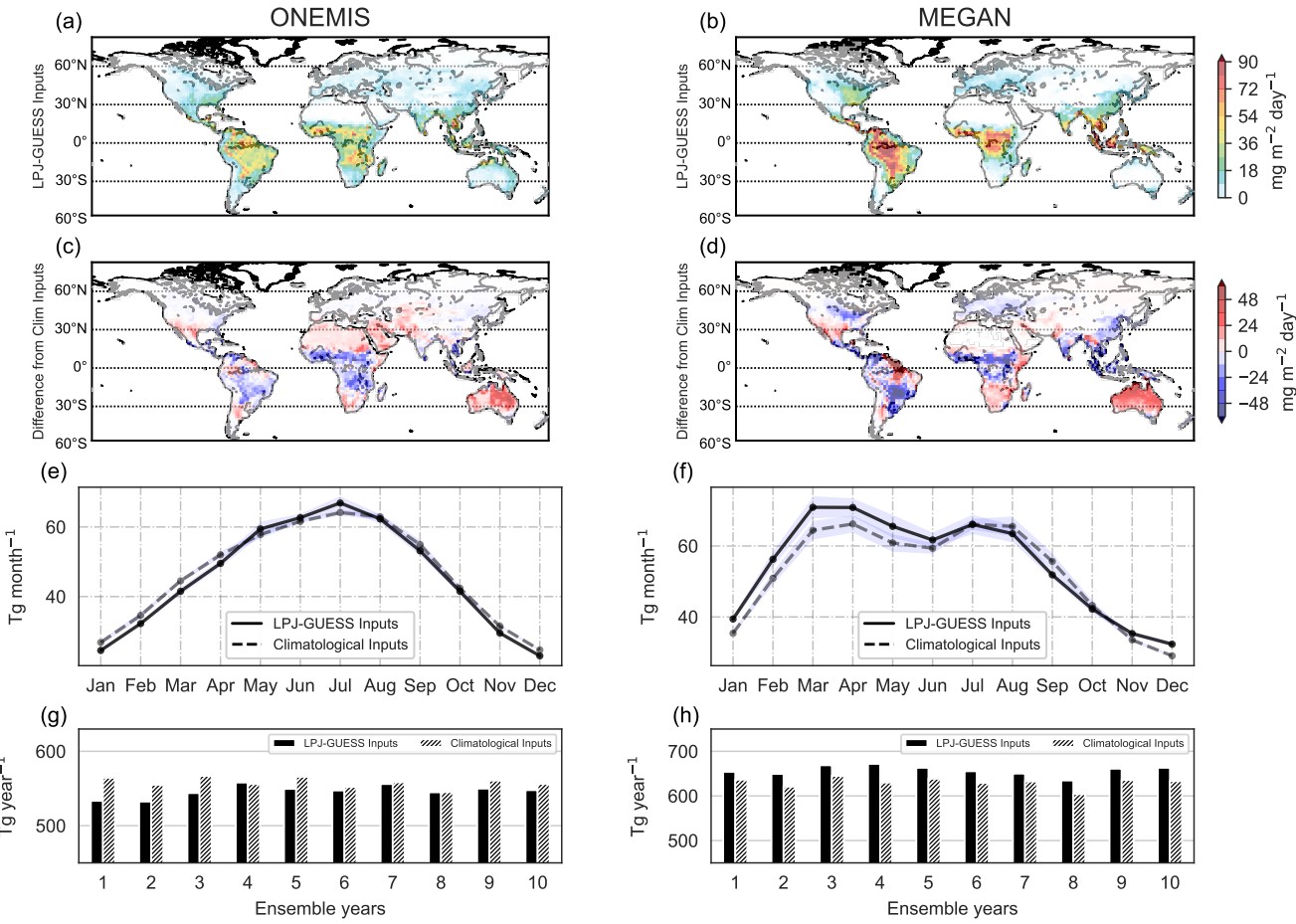

**Figure 8.** Spatial distribution of monthly isoprene emissions (mg m$^{-2}$ day$^{-1}$) averaged over 10 ensemble years with LPJ-GUESS vegetation inputs from ONEMIS and MEGAN (panels a and b). Panels c and d show the difference in the emission flux compared to emissions from ONEMIS and GUESS with climatological vegetation inputs. The temporal profile of isoprene global totals in Tg month$^{-1}$ are depicted in panels e and f, and global annual totals (Tg yr$^{-1}$) for 10 ensemble years are shown in panels g and h.

Panels e and f depict the temporal profile of global monthly emission totals. In order to capture the true seasonal cycle, values in the Southern hemisphere were shifted by six months before adding fluxes from both hemispheres. Panels g and h show the inter-annual variability of isoprene annual global totals. Over the ten ensemble years, isoprene emissions appear to peak in the boreal summer months and decrease substantially in the boreal winter months. MEGAN includes a leaf age factor which accounts for reduced emissions for young and old leaves based on observed LAI change. This explains the slight decrease in
MEGAN emissions from April to May to June.

    Over the 10-year simulation period considered, the global annual total isoprene fluxes from ONEMIS were found to be 546 Tg yr$^{-1}$ (standard deviation (SD) = 8 Tg yr$^{-1}$) with dynamic vegetation and, 558 Tg yr$^{-1}$ (SD = 7 Tg yr$^{-1}$) with climatological inputs. MEGAN estimated 657 Tg yr$^{-1}$ (SD = 11 Tg yr$^{-1}$) with dynamic vegetation and 631 Tg yr$^{-1}$ (SD = 11

Tg yr$^{-1}$) with climatological vegetation inputs. The higher standard deviation when using LPJ-GUESS inputs indicates higher interannual variability in isoprene fluxes. While the year-to-year variability with climatological inputs is only determined by surface temperature and short-wave radiation (see Fig.2), the interannual variability with LPJ-GUESS inputs is also influenced by interannual changes in vegetation states estimated in LPJ-GUESS. Jöckel et al. (2016) reported isoprene annual emissions of 488-624 Tg using ONEMIS, while other studies estimated fluxes of 642 Tg yr$^{-1}$ (Shim et al., 2005) using 73 prescribed vegetation types, 571 Tg yr$^{-1}$ (Guenther et al., 2006) using inventories and Olson ecoregions land covers, 467 Tg yr$^{-1}$ (Arneth et al., 2007a) using 10 PFTs from LPJ-GUESS, and more recently, 594 Tg yr$^{-1}$ using 16 PFTs (Sindelarova et al., 2014). It is important to note that no scaling factors were applied in our simulations, even though global annual totals from models are often scaled in atmospheric chemistry studies. For example, Pozzer et al. (2022) estimated 464 Tg of isoprene in the year 2010 using a MEGAN-EMAC setup but employed a global scaling factor of 0.6. This means that the original emissions are similar to our MEGAN fluxes with climatological inputs. In general, our LPJ-GUESS-driven BVOC emissions in EMAC agree with past modelling estimates with similar spatial and temporal patterns.

**Monoterpene emissions**

Results for monoterpene emissions are shown in Fig. 9. Panels a and b present the spatial distribution of monoterpene emission rates in mg m$^{-2}$ day$^{-1}$ from ONEMIS and MEGAN, while the panels c and d show the difference in the emission fluxes using climatological vegetation inputs. Elevated emission rates are also found over tropical rainforest regions, with ONEMIS prescribing much higher emission rates compared to MEGAN. The difference plots indicate that monoterpene emissions from ONEMIS are significantly higher with climatological inputs except for some areas over Southern Brazil and Central Africa. Similarly, MEGAN generally prescribes higher emissions with climatological vegetation inputs compared to LPJ-GUESS inputs. ONEMIS emissions over deserted regions in north Africa and central Australia appear to be better resolved with dynamic vegetation since ONEMIS with climatology inputs still attributes significant emission rates over such regions where vegetation is absent.

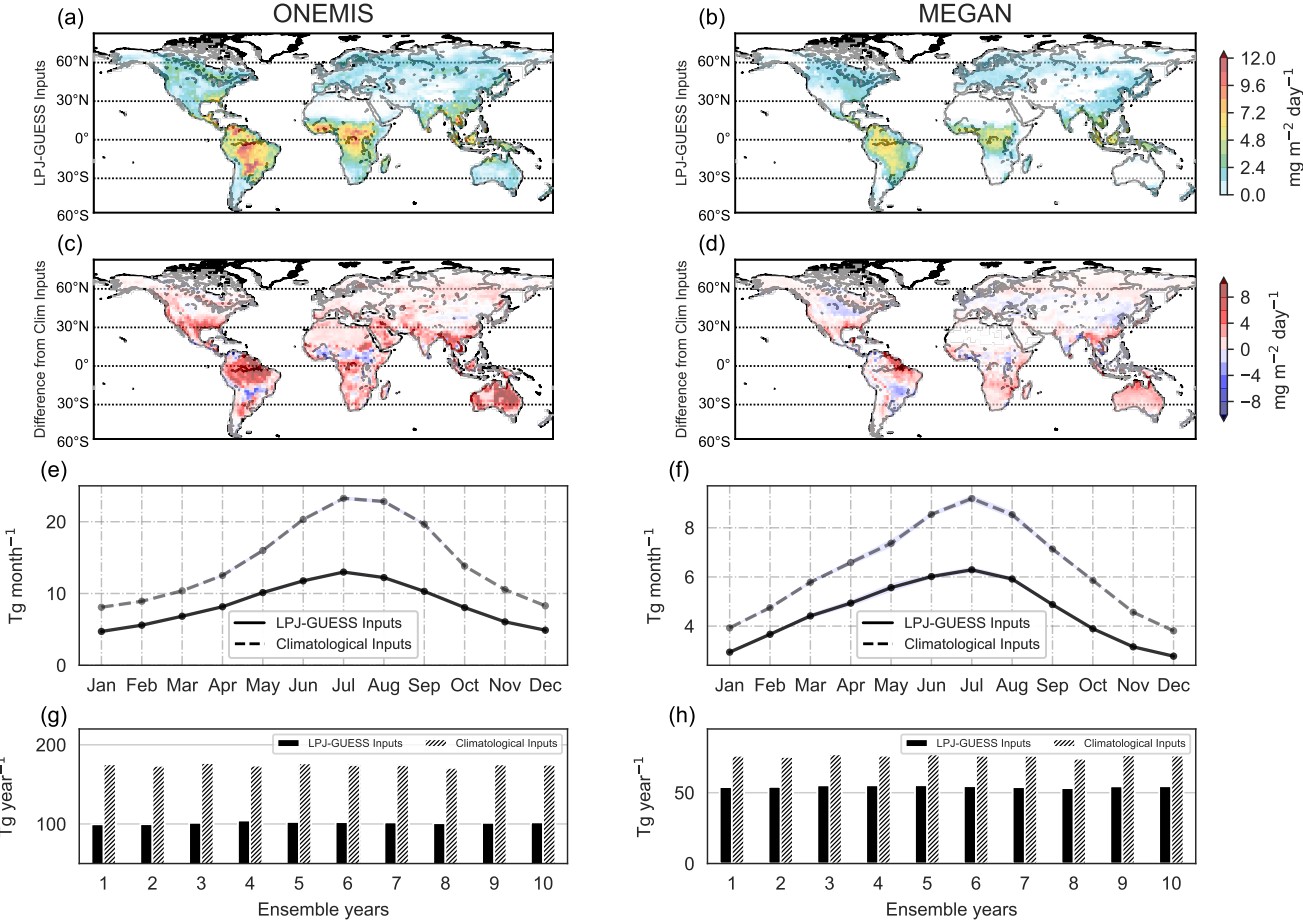

**Figure 9.** Same as Fig. 8 but for monoterpenes.

Monoterpene emission fluxes in ONEMIS only depend on the foliar density profile (i.e. DM × LAD) and surface temperature. The high dependence of monoterpene emissions on foliar density explains both the lower seasonal variability as well as the lower yearly fluxes compared to emissions with climatological vegetation inputs. The lower magnitudes in monoterpene fluxes from MEGAN with dynamic vegetation result from the lack of representation of shrubs and needleleaf tree PFTs in

LPJ-GUESS, where such species are known to be strong emitters of monoterpenes. The seasonal variation, however, is satisfying, and the fact that the fractional coverages computed in LPJ-GUESS are dynamic (updating on a yearly time step) makes this setup suitable for studying long-term variations in emissions. Annual totals from ONEMIS were found to be 102 Tg yr$^{-1}$ (SD = 1 Tg yr$^{-1}$) with dynamic vegetation inputs and 175 Tg yr$^{-1}$ (SD = 2 Tg yr$^{-1}$) with climatological inputs. MEGAN prescribes 54 Tg yr$^{-1}$ (SD = 0.7 Tg yr$^{-1}$) and 76 Tg yr$^{-1}$ (SD = 0.9 Tg yr$^{-1}$) with dynamic and climatological inputs respec-

tively. Guenther et al. (2012) gives global annual monoterpene emission of 157 Tg, while Sindelarova et al. (2014) reported

annual total emissions of monoterpenes to range between 89 and 102 Tg yr$^{-1}$ over a 30-year simulation period. Arneth et al. (2007a) reported 36 Tg yr$^{-1}$.

**BVOC emissions from LPJ-GUESS**

This section presents isoprene and monoterpene emission fluxes from the semi-processed-based module in LPJ-GUESS for
comparison with the empirical-based emissions from ONEMIS and MEGAN in the coupled model system. The LPJ-GUESS routine runs entirely within the LPJ-GUESS framework, meaning that emission values are provided on a daily basis. In this study, the LPJ-GUESS BVOC routine uses DTR computed in EMAC instead of climatological DTR (see section 2.3). Panels a and b in Fig. 10 show the spatial distribution of isoprene and monoterpene emission rates from the LPJ-GUESS BVOC emissions routine. Panels c and d show the monthly total emissions from LPJ-GUESS as well as emissions from ONEMIS and
MEGAN for comparison.

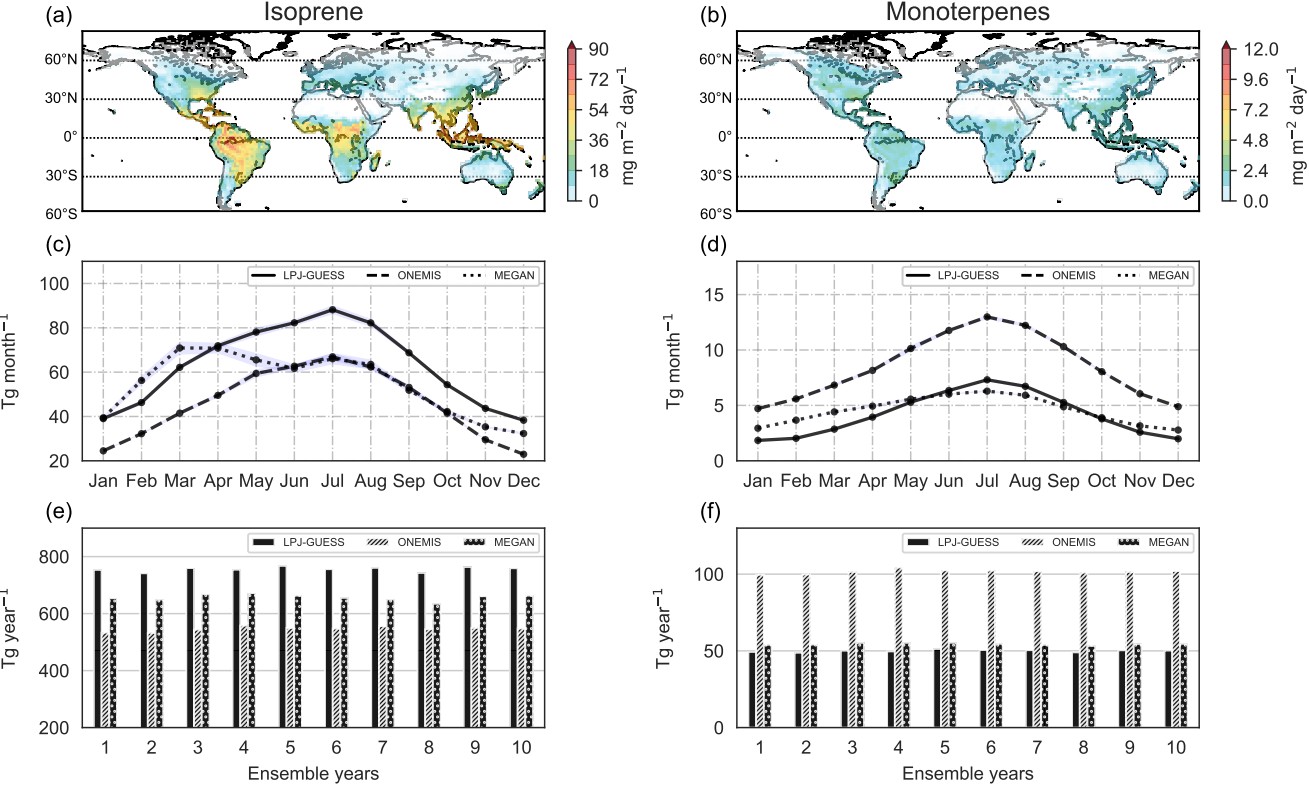

**Figure 10.** Spatial distribution of isoprene and monoterpene emissions from LPJ-GUESS at a spatial resolution of T63 (panels a and b). Isoprene and monoterpene mean monthly totals in Tg month$^{-1}$ from LPJ-GUESS, ONEMIS and MEGAN (panels c and d), and annual totals in Tg yr$^{-1}$ for 10 ensemble years (panels e and f).

Monthly isoprene emissions from LPJ-GUESS range from 48.3 Tg in December to 88.2 Tg in July, whereas monthly monoterpene emissions range from 1.8 Tg in January to 7.3 Tg in July. Panels e and f show yearly isoprene and monoterpene

totals from LPJ-GUESS, ONEMIS and MEGAN for 10 ensemble years. The mean isoprene annual emission flux is 750 Tg yr$^{-1}$ (SD = 17 Tg yr$^{-1}$) while for monoterpenes it is 50 Tg yr$^{-1}$ (SD = 1 Tg yr$^{-1}$). The process-based isoprene emissions from LPJ-GUESS are marginally higher compared to the empirical-based emissions from ONEMIS and MEGAN. For monoterpenes, emissions from LPJ-GUESS are similar to MEGAN emissions, while ONEMIS emissions are twice as much.

Currently, both empirical (e.g. Guenther et al., 2006) and process-based models (e.g. Niinemets et al., 1999; Bäck et al., 2005; Arneth et al., 2007b; Schurgers et al., 2009) are widely used to model and asses global BVOC emissions estimates. Process-based models consider emission inhibition from carbon and energy availability, allowing for some stress effects to be taken into consideration. In general, empirical-based models suggest increased BVOC emissions in future climate scenarios resulting from temperature sensitivity and enhanced vegetation activity in a $CO_2$-richer atmosphere (Sanderson et al., 2003; Naik et al., 2004; Lathiere et al., 2005). In contrast, process-based models indicate that $CO_2$-inhibition of leaf-isoprene metabolism can be large enough to offset increases in emissions (Arneth et al., 2007b; Heald et al., 2009). More recent laboratory studies provide evidence that certain plant species emit more isoprene in high $CO_2$ environments (e.g. Sun et al., 2013), indicating that there are still some gaps in the understanding of biochemical regulation of BVOC leaf emissions incorporated in such models. Grote and Niinemets (2007) compares the two model categories and argues that in non-stressful conditions, empirical and process-based emission models predict BVOC emission dependencies on light and temperature in a similarly way. A modelling study also showed that regardless of whether $CO_2$ inhibitory effects are considered or not, temperature is the most important driver for increased isoprene emissions (Cao et al., 2021). In this study, it is shown that even though process-based emissions from LPJ-GUESS might differ (in magnitude) from empirical-based emissions in EMAC, the monthly distributions in emissions are similar (Fig. 10 panel c and d). Even though emissions from ONEMIS and MEGAN in EMAC are empirical-based, emissions in the new coupled configuration are now sensitive to vegetation changes, which is a substantial improvement over emissions with prescribed vegetation information in the previous model configuration.

### 3.3 Emission sensitivity to double $CO_2$ scenarios

In this section, the variability of global isoprene and monoterpene emission estimates in doubling $CO_2$ scenarios is investigated. In particular, the $CO_2$-fertilisation and temperature effects are evaluated by prescribing different $CO_2$ values in the radiation and vegetation schemes in the coupled model setup, as described in section 2.4. The scope here is not to develop realistic future scenarios, but rather to assess the model's sensitivity to atmospheric and vegetation $CO_2$, hence the use of doubling scenarios. Four experiments were conducted to explore the impact of doubling $CO_2$ scenarios (both in the radiation and vegetation scheme) on isoprene and monoterpene emission rates. Ref is the reference simulation, with 348 ppmv [$CO_2$] in both schemes. Table 1 lists the $CO_2$ levels prescribed for each study. The analyses shown here are based on ten years of data from 50-year simulations with constant boundary conditions and a 500-year offline spin-up phase.

| Study | $CO_2$ in radiation scheme | $CO_2$ in vegetation scheme |
|:---:|:---:|:---:|
| Ref | 348 ppmv | 348 ppmv |
| Bio×2 | 348 ppmv | 696 ppmv |
| Atm×2 | 696 ppmv | 348 ppmv |
| Both×2 | 696 ppmv | 696 ppmv |

**Table 1.** Prescribed $CO_2$ in the radiation and vegetation schemes for different studies.

**Surface temperature in 2×CO2 in the radiation scheme**

The temperature sensitivity of BVOC emissions is studied by doubling the prescribed $CO_2$ value in the radiation scheme. This results in a consistent global increase in the surface temperature of up to 4°C in conjunction with the prescribed enhanced SST and sea-ice coverage (see Appendix A).

**Vegetation response to 2×CO2 scenarios**

The LAI can be used as a marker for vegetation activity. Fig. 11 displays LAI estimates from LPJ-GUESS for the reference study, Ref, and also shows how LAI values in the other studies compare to it. Bio×2 indicates consistently increased vegetation activity when doubling the prescribed $CO_2$ in the vegetation scheme. This is the $CO_2$-fertilisation effect. The LAI in Atm×2 decreases as a result of warmer temperatures and higher losses of soil moisture (reported elsewhere, e.g. Dermody et al., 2007; Sun et al., 2015). In Both×2, with 696 ppmv [$CO_2$] in both the vegetation and radiation scheme, an overall rise in LAI compared to the Ref simulation is found, except for some regions in North America, Western Brazil and Southern Europe. In these areas, water stress from higher surface temperatures results in an overall decline of vegetation, e.g., grass species take over forested areas, partly decreasing the LAI in the region. To this end, this model setup can be used to analyse changes in BVOC emissions due to shifts in vegetation patterns in future climate scenarios.

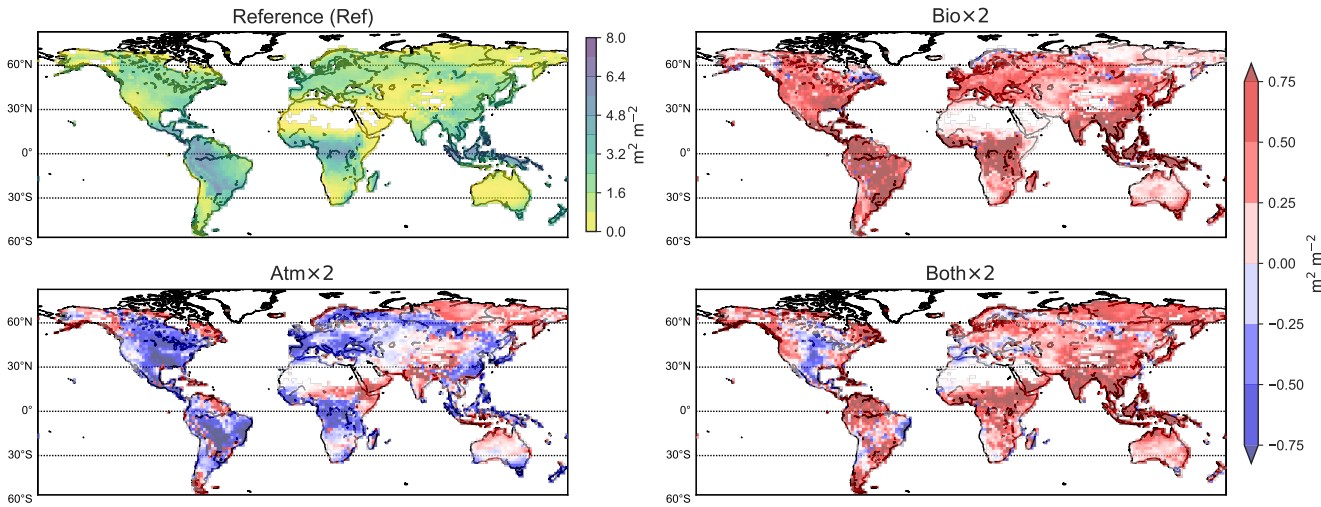

**Figure 11.** Geographic distribution of the LAI for the reference study (Ref) and difference plots; Bio×2 − Ref (top right), Atm×2 − Ref (bottom left), and Both×2 − Ref (bottom right)

### Global BVOC emissions

Fig. 12 shows the geographic distribution of isoprene emission rates (in mg m$^{-2}$ day$^{-1}$) averaged over 10 ensemble years for Ref and difference plots for Bio×2, Atm×2, and Both×2 using climatological and dynamic vegetation inputs using ONEMIS and MEGAN. The bottom panels (q, r, s, t) show the averaged emission rates for distinct latitude bands (0°- 30°S, 0°- 30°N, 30°N - 60°N) for all studies during the same period, from left to right. It is found that in Bio×2, isoprene emissions increased only when using dynamic vegetation inputs in both ONEMIS and MEGAN (panels b and j) are applied. In contrast to the prescribed climatological vegetation data, the LPJ-GUESS coupled setup is sensitive to increased $CO_2$ which subsequently leads to higher emissions. Note, that in this scenario, ONEMIS attributes lower emission values over the tropics (panel b). Isoprene emissions are more dependent on light than on leaf area, so increased foliage may limit isoprene emissions in closed canopies such as in dense tropical rain forests (Guenther et al., 2006). This is not the case for open canopies, where increased foliage drastically enhances isoprene emissions. This effect is not well-captured by MEGAN (panel j) given that here the PCEEA algorithm is employed whereas the canopy environment model only considers above-canopy radiation and is not sensitive to sun and shaded leaves at each canopy depth. In Atm×2 temperature effects on isoprene emissions are found while in Both×2 see the combined effects of $CO_2$ fertilisation and temperature become obvious. The bar plots reveal that in places where most of the global emissions occur, i.e., between 0°- 30°S, the emission was the highest in Both×2 with an average emission rate of 27.3 mg m$^{-2}$ day$^{-1}$ (ONEMIS) and 44.3 mg m$^{-2}$ day$^{-1}$ (MEGAN). Given the empirical nature of ONEMIS and MEGAN, both setups give a consistent increase in isoprene fluxes in elevated temperatures (Atm×2). However, with LPJ-GUESS inputs differences in the emission fluxes are found, also resulting from vegetation dynamics e.g., a decrease in fluxes from lower vegetation activity caused by water stresses. This highlights the advantage of having BVOC fluxes derived from dynamic vegetation states rather than prescribed boundary conditions.

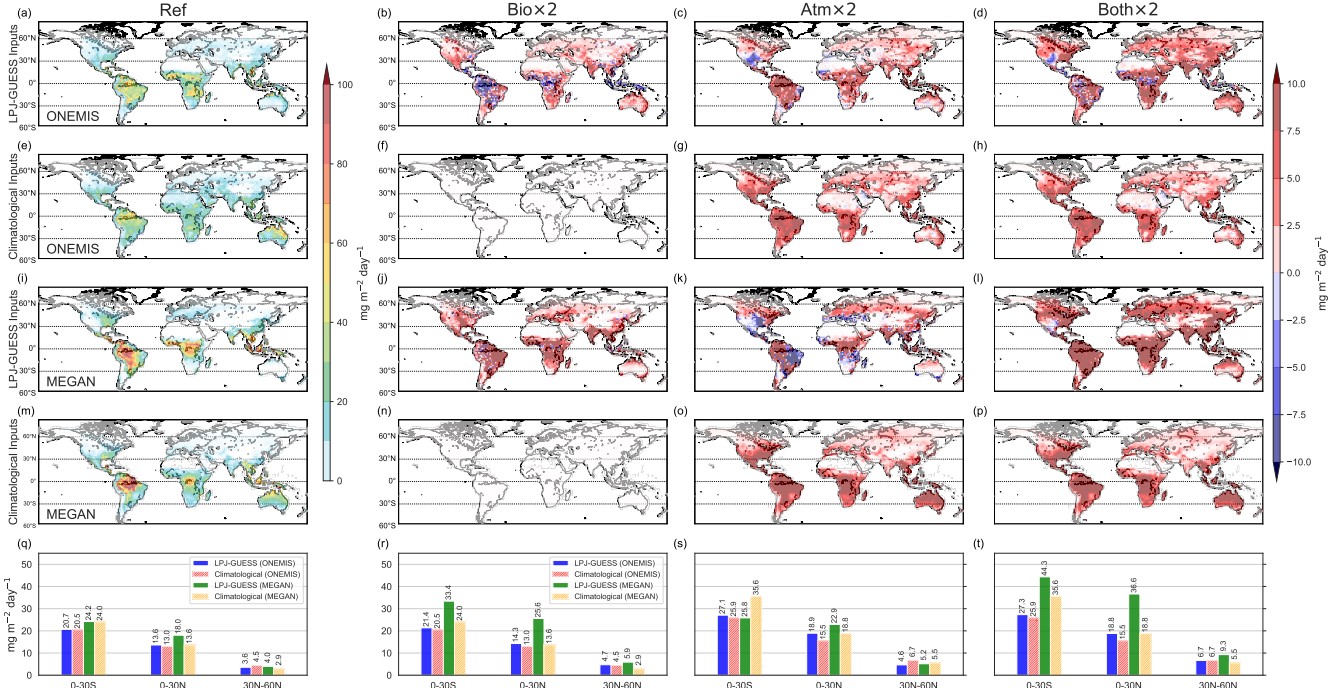

**Figure 12.** Global isoprene emission estimates from ONEMIS and MEGAN with LPJ-GUESS vegetation inputs and climatological inputs for the reference study (Ref) (panels a, e, i, m), as well as difference plots for Bio×2 (panels b, f, j, n), Atm×2 (panels c, g, k, o), and Both×2 (panels d, h, l, p). The panels in the bottom (q, r, s, t) display emissions flux averages (in mg m$^{-2}$ day$^{-1}$) over the latitude bands 0°- 30°S, 0°- 30°N, 30°N - 60°N.

In Fig. 13 similar behaviour in monoterpene emission estimates is depicted for all studies. Monoterpene emission rates increase in Bio×2 only for scenarios with dynamic vegetation due to $CO_2$-fertilisation. A worldwide increase in monoterpene fluxes in Atm×2 is detected with higher surface temperatures using climatological inputs. However, with dynamic vegetation inputs, a less substantial rise, as well as a drop in fluxes in certain regions, is determined. In Both×2 an increase in monoterpene emission rates is found as a result of the combined influence of temperature and $CO_2$-fertilisation.

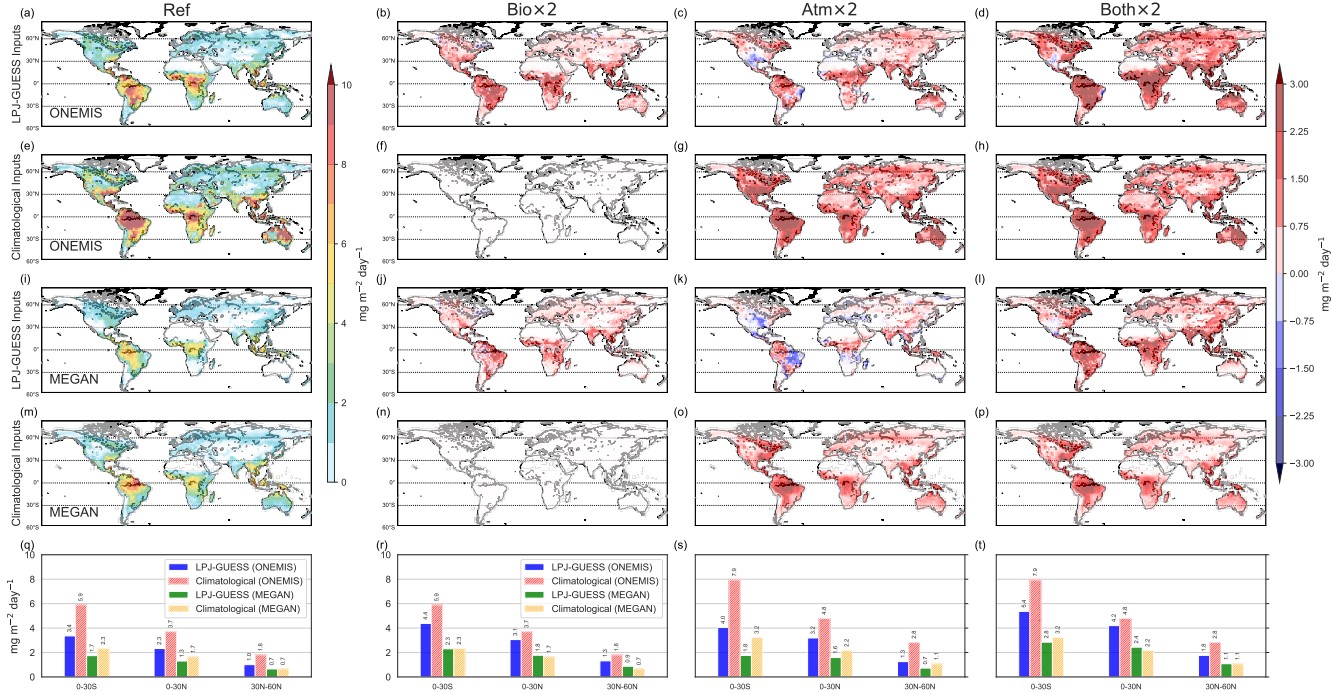

**Figure 13.** Same as Fig. 12 but for monoterpenes.

## 4 Conclusions

The development of ESMs allows a far more detailed analysis of a fully coupled and dynamic system addressing many complex biosphere-atmosphere interactions governed by BVOC emissions and thus shedding more light on the significance of such processes on global climate change and air quality. In this work, further development has been made towards producing a new atmospheric chemistry-enabled ESM by coupling an atmospheric chemistry-enabled atmosphere-ocean general circulation model (AOGCM) with a DGVM. This work also explores, for the first time, partial bi-directional interactions between the two modelling systems via BVOC emissions, building on recent technical developments (Forrest et al., 2020) that enabled one-way coupled simulations (in which climate information generated by EMAC is used to force LPJ- GUESS, but no land surface information is relayed back to EMAC). The updated model version described in this work allows computations of isoprene and monoterpene emissions based on dynamic vegetation states running on EMAC's time-step. This is a substantial improvement over the earlier setup where BVOC emissions were only based on offline vegetation information and not coupled with dynamic vegetation states. The BVOC module in LPJ-GUESS (used in this study only to compare our new emissions in EMAC, Fig. 10) uses dynamic vegetation information, but only provides daily average emissions, and these emissions are not yet integrated into EMAC.

Results show that the LAI and subsequent foliar density estimations from LPJ-GUESS are comparable to climatological datasets used as boundary layer conditions in the MEGAN-EMAC stand-alone configuration, as well as an external LAI clima-

tology from 1981 to 2015. The LAI employed in the original ONEMIS-EMAC setup, on the other hand, differs significantly in terms of magnitude and monthly variability. Such differences in the LAI inputs led to lower isoprene and monoterpene emissions using the coupled model setup compared to the stand-alone configuration. The representation of the LAD distribution from the new parametrisation employed in LPJ-GUESS also gives a more realistic LAD profile compared to the over-simplified datasets used by the standard ONEMIS setup. Given that our coupling only involves vegetative information going into ONEMIS and MEGAN in EMAC, our climate biases are considered to be consistent to those discussed in Forrest et al. (2020).

The new MEGAN-LPJ-GUESS configuration yielded satisfactory isoprene fluxes as well, with a monthly distribution comparable to MEGAN emissions with climatological inputs. Monoterpene emissions from the MEGAN-LPJ-GUESS setup also give meaningful monthly variability with lower magnitudes compared to emissions with climatological inputs given that the new setup lacks emission contributions from shrubs and needleleaf tree PFTs in mixed forests. Future studies may include region-specific PFT groups (including shrub PFTs), however, for this study, the global PFT set is kept unchanged. Global isoprene emission estimates from the coupled model configuration, using both ONEMIS and MEGAN, give realistic global variability, corresponding to emissions with prescribed vegetation boundary conditions. The emission magnitudes are also comparable to modelled fluxes found in literature and can be adjusted accordingly with global scaling factors for specific atmospheric chemistry studies. The new vegetation-sensitive emissions from ONEMIS and MEGAN agree with previous approaches with respect to the spatial and temporal patterns, but provide more consistency and higher temporal resolution as required for interactive atmospheric chemistry simulations.

This study finds that: (1) differences in BVOC emissions with implemented dynamic vegetation result almost entirely from changes in the input LAI; (2) dynamic vegetation increase the interannual variability in BVOC emissions by introducing new variability from dynamic vegetation states. These findings are comparable to those of Levis et al. (2003), where dynamic vegetation was implemented in the Community Climate System Model (CCSM).

$CO_2$ sensitivity studies also suggest that both isoprene and monoterpene emissions rise with warmer climatic scenarios ($2 \times CO2$ in the radiation scheme), however only the coupled configuration showed reduced emissions in some locations. The lower emissions result from the vegetation response to water stresses in a warmer climate. The new coupled model also allows for sensitivity studies for $CO_2$-fertilisation effects and indicates an increase in both isoprene and monoterpene emission rates in $2 \times CO2$ scenarios in the vegetation scheme due to enhanced vegetation activity in a $CO_2$ rich-atmosphere. This work provides evidence that the improved ESM, featuring dynamic vegetation, gives realistic BVOC emission estimates on a global scale based on dynamic vegetation states, enabling further research into atmosphere-biosphere interactions and feedbacks with this model configuration.

*Code availability.* The Modular Earth Submodel System (MESSy) is continuously developed and applied by a consortium of institutions. MESSy is licensed to all affiliates of institutions that are members of the MESSy Consortium, as is access to the source code. Institutions can become a member of the MESSy Consortium by signing the MESSy Memorandum of Understanding. More information can be found on the MESSy Consortium website (http://www.messy-interface.org, last access: 14 January 2023). LPJ-GUESS is used and developed globally,

however, development is overseen at Lund University's Department of Physical Geography and Ecosystem Science in Sweden. Model codes can be made available to collaborators on entering into a collaboration agreement with the acceptance of certain conditions. Given that both MESSy and LPJ-GUESS are restricted, the code used here is archived with a restricted access DOI (https://doi.org/10.5281/zenodo.6772205). The code will not be made publicly available and sharing will be made possible only by the approval of the authors. The code described in
this manuscript has already been incorporated into the official development branch of the EMAC modelling system and will therefore be part of all future released versions.

## Appendix A:  Surface temperature at doubling CO$_2$ scenarios

Fig. A1 illustrates temperatures ($^\circ$C) at the lowermost vertical level in EMAC at spatial resolution T63 with prescribed CO$_2$ of 348 ppmv in the top panel and 696 ppmv in the middle panel. A plot comparing the reference simulations to the 2×CO2
simulations is given in the lower panel of Fig. A1.

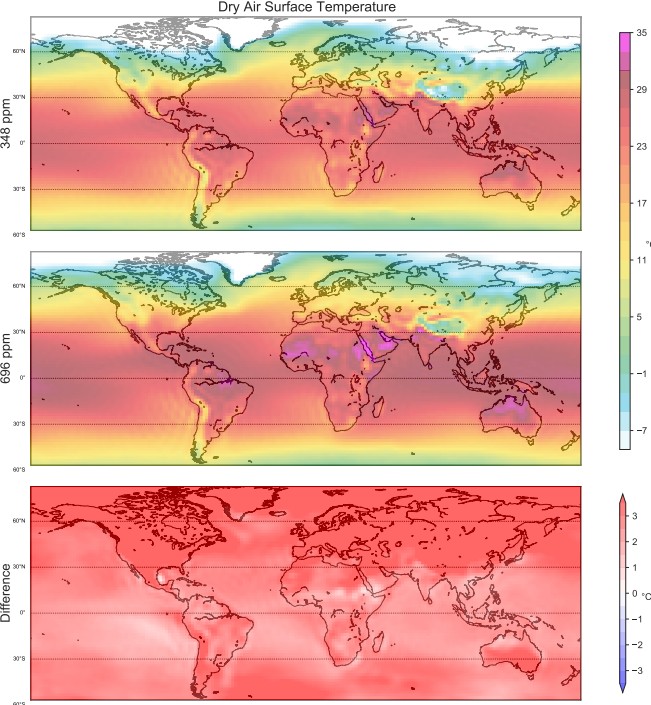

**Figure A1.** Surface temperature ($^\circ$C) at spatial resolutions T63. In the radiation scheme, the first model configuration utilises a CO$_2$ volume mixing ratio of 348 ppm, whereas the second setup uses 696 ppm.

*Author contributions.*  RV and HT performed the model coupling. RV performed the simulations and analysis. All authors contributed to the overall model development, scientific analysis and writing of the article.

*Competing interests.* HT acts as a topical editor for GMD. Apart from this, the authors declare that they have no conflict of interest.

*Acknowledgements.* This research was conducted using the supercomputer Mogon and/or advisory services offered by Johannes Gutenberg
University Mainz (https://hpc.uni-mainz.de/, last access: 14 January 2023), which is a member of the AHRP (Alliance for High Performance
Computing in Rhineland Palatinate, https: //www.ahrp.info/, last access: 14 January 2023) and the Gauss Alliance e.V. RV thank Andrea
Pozzer (Max Planck Institute for Chemistry, Mainz) for technical support to implement MEGAN in the new model configuration. This work
was supported by the Max Planck Graduate Center with the Johannes Gutenberg-Universität Mainz (MPGC). We also acknowledge funding
from the Carl-Zeiss foundation for HT. Finally, we thank four anonymous reviewers and the handling editor for their helpful comments.

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
