# Peer review of "Isoprene and monoterpene simulations using the chemistry-climate model EMAC (v2.55) with interactive vegetation from LPJ-GUESS (v4.0)"

_Geoscientific Model Development, 2022_

## Author Comment (AC1)

We thank the referees for taking the time to review our manuscript and for the valuable feedback. We have corrected our manuscript according to the referees' comments and think it is now significantly improved. Please find here our point-by-point response to the referees' comments.

*The referee's comments are in blue while our responses are in black.*

**Anonymous Referee #1**

This work gives very nice information on the recent development of global-scale BVOCs emissions modeling with multi-model and components interactions. And there are authors' efforts to evaluate the results with a lot of previous works including scientific reviews. I think this research shows an advanced way of estimating BVOCs with more realistic interactions with Earth components. I have only a few questions and suggestions for the publication as follows.

We would like to thank the referee for the positive feedback and the recommendation for publication.

**Remarks:**

1) Figure 7 and 8, and 9: Please make the same y-scale both at the ONEMIS and MEGAN results (middle and bottom panels), such as in Figure 11.

   We updated the figures with the same colour-bar scales.

2) 3.3. I understand the author wanted to measure the sensitivity of doubling atmospheric CO2 and vegetational CO2 separately. However, the increasing CO2 influences the vegetational CO2 in reality. The author should mention about "real" future conditions or add the case with scenarios of realistic future conditions of Bio and Atm.

   We provide three further scenarios to compare with our standard case: one with doubling CO2 in the vegetation scheme, one with doubling CO2 in the radiation scheme, and both. The "Both x 2" case would be the more realistic scenario here but this exercise aimed to test the sensitivity of the coupled model rather than making realistic future predictions. The manuscript was modified to clarify better the aims of this section.

3) Line: 310: Why does the author think that Both x 2 scenarios showed some exceptions of lower LAI over some places in North America, Western Brazil, and Southern Europe? That needs a few scientific explanations like partially described in conclusions.

In Both x 2 scenarios, the LAI (i.e. vegetation growth) is influenced both by an increase in vegetational $CO_2$ (fertilisation effect), as well as increased surface temperatures from increased $CO_2$ in the radiation scheme. The decrease in LAI in the central USA, some parts of South America, and southern Europe results from competition between plant species resulting in shifts in vegetation between PFTs, e.g. grass species take over forested areas, partly decreasing the LAI. These shifts are mostly dominated by water stresses. The text was modified accordingly but we think providing too much detail here is beyond the scope of this paper. However, we indeed want to evaluate such vegetation shifts in future studies.

**Anonymous Referee #2**

"Isoprene and monoterpene simulations using the chemistry-climate model EMAC (v2.55) with interactive vegetation from LPJ-GUESS (v4.0)", by Ryan Vella et al.

The manuscript by Vella et al. documents the coupling between EMAC and two terpenoid emission schemes driven by dynamic vegetation from LPJ-GUESS. The results from the two schemes are compared against each other, as well as to emissions computed directly inside LPJ-GUESS. In addition, a series of doubled $CO_2$ concentration simulations was performed to illustrate the sensitivities of the simulated terpenoid emissions to these.

Overall, the paper presents an important linkage between ecosystems and atmospheric chemistry, and it is good to see this linkage represented in the EMAC system. While the simulations themselves and their analysis may not be very novel, the implementation serves a clear purpose, and a manuscript like this documenting model development is well suited for GMD.

The description of the coupling may need some clarifications (see below) but is overall fine, and analysis and its description are understandable, but the manuscript would benefit from a better explanation of the simulation setup in the Methods section, to provide the information on the simulations on beforehand. In particular, it would be good to describe whether these simulations are run as bi-directional interactions (L. 336) including changes in the climate caused by the changes in atmospheric chemistry, or whether the setup simply tests the emission response, but not the EMAC response to these (which I guess is the

case). Also, it would help to understand the role of the results from the BVOC emission routine in LPJ-GUESS (Fig. 9), which are presented separately from those of the two other schemes – are these available for use in EMAC as well, or are they only presented here for comparison?

Apart from that, it would be good to clarify more clearly in the Methods section which elements of the coupling come from LPJ-GUESS, and which are assumptions that are used to "interpret" the LPJ-GUESS results inside ONEMIS or MEGAN in the model description. I have indicated the places that are unclear below.

I expect that the manuscript will be suited for publication in GMD once these comments have been accounted for.

We thank the referee for the feedback and suggestions to improve our manuscript, as well as the recommendation for publication. Detailed responses are below.

**Major remarks:**

1) L. 23: Oxidative stress is one possible reason for BVOC emissions, but they can also be triggered by other chemical, physical or biological stresses and processes (e.g. herbivory, signaling between organisms, or also oxidative stress originating from the atmosphere, e.g. under elevated ozone concentrations.

We have modified the manuscript accordingly.

2) L. 25: I think that all plants emit BVOC, but they can emit very different compounds, and not all emit isoprene.

This is correct. The text was modified.

3) L. 91: It would be interesting to summarize the difference between ONEMIS and MEGAN a bit further. E.g., in the later text, it appears that the two treat canopy structure in a different way. It would be good if the authors could give a brief description of the two schemes, as they are so fundamental for the rest of the paper.

We included a table summarising the key differences between ONEMIS and MEGAN (Table1). The cited papers should provide all details about the algorithms.

4) L. 107: I think that LPJ-GUESS v4.0 contains a functional land use scheme, see Lindeskog et al. 2013. In general, I think that the fact that land use is not represented should receive more attention in the discussion, in particular because the original emission schemes appear to represent crops. How important is this omission for the outcomes generated by the LPJ-GUESS-informed emissions schemes (ONEMIS and MEGAN)?

Even though LPJ-GUESS v4.0 contains a land use scheme, the EMAC-LPJ-GUESS configuration has no land use implemented yet (Forrest et al., 2020 for reference). In the discussion section, we further emphasised that our emissions are from the natural biosphere. We are aware that this is a limitation, but it is good to note that climatological values for LAI also do not fully take managed land into account. Nevertheless, the impact of crops is going to modify the emission results to a certain degree - manuscript updated.

5) Section 2.3.2: The authors have chosen to use LPJ-GUESS to provide information on LAI and PFT distribution, but other characteristics that are required for ONEMIS or MEGAN are not taken from LPJ-GUESS, but rather computed with the help of database numbers.

This is not entirely correct. As explained in section 2.3.1, all vegetation variables (i.e. LAI, DM, LAD distribution for ONEMIS; and LAI, vegetation-type coverage for MEGAN) are taken from LPJ-GUESS. The BVOC output from ONEMIS and MEGAN is then compared to the "original" set-up, where all vegetation variables are taken from database numbers. In the EMAC-LPJ-GUESS configuration, the only characteristics required for ONEMIS and MEGAN that are not taken from LPJ-GUESS are (1) emission factors, (2) surface temperature, (3) short wave radiation, and (4) solar zenith angle.

Foliar density (L. 135) is computed from simulated LAI, rather than from the foliar C simulated by LPJ-GUESS. Why is this done? And how similar or different are the applied specific leaf weights from those used in LPJ-GUESS itself?

This is a good point, however, we opted to compute the foliar density (DM) as described in the papers for two reasons: (1) The use of S_LW values from Olson makes our estimations consistent with our emissions factors and the framework of the calculations, (2) the leaf mass (cmass_leaf) from LPJ-GUESS was not available on the EMAC side.

In LPJ-GUESS the cmass_leaf could be calculated as follows:

cmass_leaf = LAI/SLA

where SLA is the specific leaf area per PFT. LPJ-GUESS calculates the SLA from the leaf longevity and leaf physiognomy values for each PFT.

We compared our foliar density with the foliar C simulated by LPJ-GUESS (Fig. 1) and the differences aren't large. We think that for the purpose of this study, and given that our DM and the one from LPJ-GUESS are so similar, the presented values are sufficient. We understand that in future studies when land use is implemented, there might be differences in the spatial patterns when compared to observations. In that case, we would highly consider using the cmass_leaf directly from LPJ-GUESS.

[Figure]

Figure 1

The same applies to the LAD distribution, which is taken from some standardized profiles, rather than using LPJ-GUESS' vertical distribution of LAI. It would be nice to hear more about this, and mention explicitly which information comes from LPJ-GUESS, and which from other (literature) sources. E.g., I guess that the canopy height (h) in Eq. 2 comes from LPJ-GUESS and does not use the fixed height of 25 m (L. 160), given the simulated variations in canopy height (Fig. 5), but I cannot find this in the description.

For the LAD distribution calculation, the PFTs are categorised into three vegetational types as previously done in ONEMIS using a 21-layer canopy DDIM point data. The improvement here is that the parametrisation employed allows for the LAD distribution to be calculated as a continuous function. We still use the PFT height from LPJ-GUESS to evaluate the LAD distribution of each PFT within an assumed canopy height of 25m.
We understand that this might not have been very clear. The text is now modified

accordingly. We also make the distinction between PFT height ($h$) and total canopy height ($h_{tot}$).

6) Results: At several places, seasonal variations are displayed as global mean (Fig. 4 bottom panel; Fig. 7 and 8 third row, Fig. 9 second row). However, the opposite seasons in the Northern and Southern hemisphere make it hard to interpret these; it would be nice to see them separated for the two hemispheres, or have them shifted by 6 months before adding, to represent the true seasonal cycle.

We thank the reviewer for pointing this out. All figures showing the seasonal variation were updated. Values from the southern hemisphere were shifted by 6 months as suggested.

7) L. 225: I agree that the representation of LAI has improved, but for the isoprene emissions, it is also important that the vegetation distribution has improved. Is this the case? See also my earlier remark on the representation of crops.

In Forrest et al. 2020, it has been already shown that the vegetation distribution in reasonable in the coupled model. Even in case of worse agreement compared to the offline data, the consistency in the new setup is higher, such that feedback mechanisms can be investigated.

8) L. 267: Why is the BVOC emission routine from LPJ-GUESS presented separately here? Is it available for use in EMAC, or is it just for comparison here? It would be interesting to see the results compared to Fig. 7 and 8 (again, please ensure that colour scales are the same). The description of the BVOC emission routine in LPJ-GUESS should be part of the methods section – this would also help to clarify what the status of this is relative to the other two emission schemes.

BVOC emissions from LPJ-GUESS are presented for comparison only. Fig. 9 now has the same colour-bar scales as Fig. 7 and Fig. 8, and panels (c), (d), (e), and (f) also include data from ONEMIS and MEGAN. The description of the LPJ-GUESS routine was moved to the methods section.

The LPJ-GUESS routine runs entirely on the LPJ-GUESS side. One major difference is that the EMAC routines (ONEMIS and MEGAN) run on short time step values (according to the model's time step e.g. 10 minutes), while LPJ-GUESS only give daily emission values. The manuscript was updated accordingly.

9) Section 3.3: The description of the setup of the sensitivity simulations should be part of the methods section.

The description moved to the methods section.

**Minor remarks:**

1) Figures: It would help to add labels (a, b, etc.) to the panels in the figures, to make the references to the figures more accurate.

Figures now include labels.

2) L. 37: Check spelling of "monoterpene"

Corrected.

3) L. 48: Not all stress effects are represented (properly) in our current process-based models.

Text updated.

4) L. 90: "this schemes" – does this refer to the two emission modules?

Yes. Text updated for clarification.

5) L. 132: "number of leaves": Do you mean "amount of leaves"?

Yes. Text updated.

6) L. 202: Clarify that "broadleaf" and "needleleaf" are trees.

Text updated.

7) Fig. 7 and following figures: Please ensure that the colour scales for the ONEMIS and MEGAN panels are the same, so that the patterns can be easily compared. Also, please clarify the use of the "climatological input": Is this a climatological input to ONEMIS and MEGAN (and how do the two schemes compare when running these climatological results), or is this one set of climatological input of emissions to

Figures caption slightly updated. "climatological input" indeed refers to vegetation inputs to ONEMIS and MEGAN from offline climatology datasets. We think this term is well defined in section 2.3.1 including Fig. 1. Albeit not compared in the same plot, ONEMIS and MEGAN emissions using climatological inputs are included in Fig. 7 (and following figures) in panels (e) and (f).

8) L. 213: "Elevated" isoprene emissions. Elevated relative to the climatological inputs? Please specify what the reference level is here.

Text updated.

9) L. 236: "cross-annual": do you mean interannual?

Yes. Text modified.

10) Fig. 11: Please check the figure quality/resolution for the final publication, it is a bit blurred in the discussion paper.

Figure resolution updated from 300 to 500 dpi.

11) L. 331: The first sentence could be removed.

Agreed.

12) L. 342: Check spelling of "climatology"

Thanks for pointing this out.

13) L. 349: "when the difference in the prescribed monthly LAI": Do you mean the year-to-year difference here?

No. Here we mean the monthly input LAI. It should be more clear now.

**Reference:**

Forrest, M., Tost, H., Lelieveld, J., and Hickler, T.: Including vegetation dynamics in an atmospheric chemistry-enabled general circulation model: linking LPJ-GUESS (v4. 0) with the EMAC modelling system (v2. 53), Geoscientific Model Development, 13, 1285–1309, 2020.

---

## Referee Report (RR1)

The revision of the manuscript has been made suitable for publication through the major revision. This study developed a geoscientific model that calculates global BVOC emissions by connecting ONEMIS and MEGAN in EMAC modules to LPJ-GUESS, and tested the sensitivity of emissions through CO2 doubling experiments.

I think that the experimental results of this study alone are scientifically meaningful findings and numbers. Here are some suggestions for minor fixes.

1. Please consider rephrasing sentences from Forrest et al., 2020, GMD
   - L90~91   and L105~108
   - If there are more sentences, ...

2. Table 1 is hard to read. How about arranging it in one or two sentences?

3. Other minor comments are below:

@ abstract
- emissions from terrestrial vegetation, which represents
   > emissions from terrestrial vegetation, which represent

- Please consider rephrasing this sentence:
   and atmospheric chemistry is a recommended tool to address the fate of
     > and atmospheric chemistry is recommended to address the fate of

- were found to be > were (delete "found to be")

- conclude that the proposed model setup is a useful tool for
   > conclude that the proposed model setup is useful for

@L35
  - the main precursor > the primary precursor

@L166
 - ecosytem > ecosystem

@L199
 - long-wave > longwave

@L265
 - climatologcial > climatological

@L340
 - water stress from higher surface temperatures result
   > water stress from higher surface temperatures results

@L355
 - where increased foliage drastically enhance isoprene emissions.
   > enhances

@L420
 - wrting > writing

@L586
 - meteorology > Meteorology

---

## Editor Decision (ED1)

I don't believe that authors properly dealt with my concerns in the previous review process. Please understand my comments are important to figure out the important aspects of this study. The authors need to revise the manuscript not to mislead readers of this manuscript. Please understand that misleading possibility comes from Introduction and Abstract mainly and must clarify what are improved from Forrest et al. (2020). Also please give us proper responses to a new reviewer who also pointed out this point. I am also concerned that we may feel a salami slice issues if the issues below are not properly dealt with.

1. This study evaluates the dynamic vegetation state simulated b LPJ-GUESS. LPJ-GUESS is one of famous biosphere models which has been improved by many independent scientists and its evaluations have been done. Eventually, it seems that this study extended Forrest et al. (2020) by coupling LPJ-GUESS to EMAC earth system model. This point should be mentioned clearly in Introduction and Abstract for better readability. The manuscript should clearly describe what this study did in abstract and introduction. Figure such as Fig. 1 in Forrest et al. (2020) is helpful to understand important aspect of this study quickly.

2. Based on the introduction, it seems to us that this study considers a semi-process BVOC emission module by Niinemets (2010) in the EMAC ESM coupling work in this study. s authors pointed out, there are already BVOC modules of ONEMIS and MEGAN in EMAC GCM. Because LPJ-GUESS has a semi-process BVOC emission module by Niinemets (2010), we generally expect that your study combines the LPJ-GUESS BVOC module into the EMAC GCM and we may want to know how such process-based BVOC module improves the simulation.

   However, this is misleading because EMAC ESM uses ONEMIS and MEGAN by series of papers by A. Guenther, not Niinements et al. (1999). Introduction should properly describe this point and must be rewritten for better understanding of improvements by this study.

   It is also important to clearly mention that the BVOC emission module (i.e., process-based model) is not used in the EMAC ESM. For example, in introduction, the manuscript mentioned a few important improvements in the LPJ-GUESS for BVOC simulations (e.g., process-based model for BVOC emission), but such process based model in LPJ-GUESS is not used in the EMAC. It makes us difficult to catch up the important works of this study.

3. Fig. 6 says that LPJ-GUESS produces no shrub in our earth which may be not true. I ask the authors to explain why there is no shrub land by LPJ-GUESS and implications for BVOC emission.

4. This manuscript compares ONEMIS and MEGAN empirical BVOC models to LPJ-GUESS module in Fig. 9 and I ask the authors to include how to calculate BVOC emission in the LPJ-GUESS.

5. The manuscript is saying that BVOC emission from ONEMIS and MEGAN is different (e.g., differences in tropical regions) but it will be better to explain (although that is simple) what makes such differences (e.g., LAI difference, temperature difference) and its implications for future climate simulations and BVOC emission uncertainties.

6. Such as Figure C1-C3 in Forrest et al. (2020), it will be better to quantify EMAC climate biases after improving the coupling processes between EMAC and LPJ-GUESS.

7. We feel that this study needs to mention previous studies to deal with dynamic vegetation model incorporated ESM (e.g., Levis et al., 2003) for clear understanding of what this study did. Also, it will be better if your work is compared to previous simulations (even though with simple or descriptive).

8.
9. Also, vegetation state such as LAI can be evaluated easily with remote sensed data such as MODIS and I am not sure if Fig. 4(d) corresponds to this reference data or not. Please provide more information on the data for Fig. 4(d) and please provide remote sensed LAI data for the reference.

---

## Author Response (AR2)

Author's response to editor comment:

**"Isoprene and monoterpene simulations using the chemistry-climate model EMAC (v2.55) with interactive vegetation from LPJ-GUESS (v4.0)"**

5  by Ryan Vella et al.

We thank the editor for taking the time to handle our paper submission and we appreciate his comments to further improve our manuscript. We would also like to thank the third referee for the time to review the revised version of our manuscript and recommending our work for final publication. Here, the editor's comment (from October 19, 2022) is reproduced in black, while our comments are presented in
10  blue.

**From the editor's response:**

Thank you very much for your valuable contribution to our journal. Despite encouraging comments from reviewers, my own evaluation suggests major revision for our journal publication as a model description paper. This is a model description paper, and the manuscript must provide clear and full information
15  on what you did for the model development and originalities of the study accordingly. But you are emphasizing the coupling of a process based-model for BVOC emission to the ESM in Intro and abstract. Based on this clear understanding, I may be able to decide if your sensitivity experiments provide relevant information for scientific progresses.

We thank the editor for the feedback and his acknowledgement of positive comments form the reviewers.
20  We understand that editor expects some clarifications about our model development efforts to make this work more fitting in the category of *"model description papers"* within the journal GMD. The manuscript was updated to clarify that we only used the built-in LPJ-GUESS "process-based" BVOC module for comparing our new emissions from ONEMIS and MEGAN with dynamic vegetation states (see Fig. 9). Emissions from the LPJ-GUESS module are not integrated in EMAC and are only limited to LPJ-GUESS'
25  daily output. Process-based models were mentioned in the introduction to give an overview of different modelling approaches. It should however be clear that in this study we utilised empirical models i.e. ONEMIS & MEGAN.

Most of all, it is hard to catch up what this study improved the BVOC emission. It is confusing if your work provides the process based-model by Niinemets and Arneth with the EMAC ESM or not. It seems

30  to me that your model development enables to provide the vegetation related parameters (e.g., LAI) from LPJ-GUESS to ONEMIS and MEGAN based on Fig. 1 (e.g., see direction of arrow to LPJ-GUESS from EMAC) and description in 2.3.1.

Fig. 1 was updated to better explain our contribution in this paper. The arrow from EMAC to LPJ-GUESS refers to the one-way coupling between EMAC and LPJ-GUESS, done by Forrest et al. (2020). In Forrest
35  et al. (2020), the atmospheric states needed in LPJ-GUESS where taken from EMAC but no vegetation information was passed back to EMAC. In this work, we replaced offline vegetation information needed in ONEMIS and MEGAN with interactive vegetation states from LPJ-GUESS, which is already coupled with EMAC. This is indeed a significant improvement in the representation of BVOC emissions in EMAC because the emissions are now vegetation-sensitive. In our sensitivity studies we showed for instance that
40  BVOC emissions are now sensitive to vegetation changes in doubling $CO_2$ scenarios. Fig. 11 and 12 show that changes in BVOC emissions in the Bio×2 case only occur in the improved coupled system.

The manuscript was updated to clarify that the process-based model by Niinemets and Arneth is only used in the LPJ-GUSS *built-in* module and is not integrated in EMAC. We used such emissions in Fig. 9 for comparison only. Now, we also highlighted that BVOC emissions from the LPJ-GUESS module are
45  limited to daily means in contrast to our new vegetation-sensitive emissions from ONEMIS and MEGAN which run on the model's time-step and thus give diurnal variations. This is one more improvement enabled by our work.

Before moving to the next step, I kindly ask you to provide full revision on the model description part so that the model communities can get benefits from your study by easy understanding what you did. Here
50  I did not give specific comments but please also make sure that all abbreviations are provided properly in the texts. Also please give us more specific words instead of "some", "the sensitivity"(of what), "the functionality"(of what), "limited two-way" (limited?).

We agree that the editor raised valid concerns about our model description section and the feedback provided improved the quality of our manuscript. We think that this version of the manuscript further
55  clarifies our approach and model development efforts to advance the integration of the EMAC ESM with LPJ-GUESS. We hope that the updated model description section addresses the editor's concerns.

We also updated our manuscript accordingly making sure that all abbreviations are explained in the text the first time they appear and we also provide more precise descriptions replacing any ambiguous remarks.

---

## Author Response (AR3)

Author's response to editor comment (second round):

**"Isoprene and monoterpene simulations using the chemistry-climate model EMAC (v2.55) with interactive vegetation from LPJ-GUESS (v4.0)"**

5   by Ryan Vella et al.

We thank the editor once more for taking the time to handle our paper submission and we appreciate the comments to further improve our manuscript. We understand that we may have not fully addressed the editor's concerns in the last round of revisions and we thank the editor for providing further details on the matter. Here, the editor's comment (from November 25, 2022) is reproduced in black, while our

10  comments are presented in blue. Further down in this document, we also address the comments from the fourth referee. We thank again all anonymous referees for their valuable contributions.

**From the editor's response:**

I don't believe that authors properly dealt with my concerns in the previous review process. Please understand my comments are important to figure out the important aspects of this study. The authors need

15  to revise the manuscript not to mislead readers of this manuscript. Please understand that misleading possibility comes from Introduction and Abstract mainly and must clarify what are improved from Forrest et al. (2020). Also please give us proper responses to a new reviewer who also pointed out this point. I am also concerned that we may feel a salami slice issues if the issues below are not properly dealt with.

20  The above concerns are tackled in detail below.

(1) This study evaluates the dynamic vegetation state simulated b LPJ-GUESS. LPJ-GUESS is one of famous biosphere models which has been improved by many independent scientists and its evaluations have been done. Eventually, it seems that this study extended (Forrest et al., 2020) by coupling LPJ-GUESS to EMAC earth system model. This point should be mentioned clearly in Introduction and

25  Abstract for better readability. The manuscript should clearly describe what this study did in abstract and introduction. Figure such as Fig. 1 in Forrest et al. (2020) is helpful to understand important aspect of this study quickly.

As stated, this study extends on Forrest et al. (2020) by further coupling LPJ-GUESS to EMAC. We enable vegetation-driven emissions in EMAC using LPJ-GUESS information. The abstract and introduction were updated based on this feedback. We reproduced the original roadmap of the model coupling strategy between EMAC and LPJ-GUESS (Fig. 1 in Forrest et al., 2020). Our new figure (Fig. 1 in the updated manuscript), clearly highlights the improvements, new developments, and how this study fits in the current and planned implementations to tighten the coupling between EMAC and LPJ-GUESS.

(2) Based on the introduction, it seems to us that this study considers a semi-process BVOC emission module by Niinemets (2010) in the EMAC ESM coupling work in this study. s authors pointed out, there are already BVOC modules of ONEMIS and MEGAN in EMAC GCM. Because LPJ-GUESS has a semi-process BVOC emission module by Niinemets (2010), we generally expect that your study combines the LPJ-GUESS BVOC module into the EMAC GCM and we may want to know how such process-based BVOC module improves the simulation.

The introduction gives a general overview of the current literature on modelling BVOC emissions, where we mention both empirical and process-based approaches. Few sentences were added (L. 53 & L. 59) to emphasise that we do not consider semi-process based BVOC emission modules in EMAC. The semi-process based module in LPJ-GUESS is explained in Section 2.2 ("BVOC emission routine"). Given that we only use emissions from this module to compare our new emissions from ONEMIS and MEGAN in EMAC, we think that the current description is sufficient. The reader could refer to the cited litterateur for technical details on the algorithm.

The implementation of the full emission scheme of LPJ-GUESS goes beyond the scope of this study. The LPJ-GUESS emission scheme has been designed to operate on at least daily time steps. An adaptation to the shorter (a few minutes) time step of EMAC is rather complicated, especially, when the current scheme uses daily average light fluxes and a daily temperature range instead of individual snapshots of radiative fluxes and temperature. This would require a complete re-tuning of the emission scheme, with the only benefit of the higher temporal resolution of the emission fluxes (which cannot be utilised in LPJ-GUESS, but in EMAC only). Even though the scheme of Niinements is semi-process based, the processes are also highly parameterised, such that the advantages against the Guenther et al. algorithms are also small. This information has been added in the manuscript (L. 140)

However, this is misleading because EMAC ESM uses ONEMIS and MEGAN by series of papers by A. Guenther, not Niinements et al. (1999). Introduction should properly describe this point and must be rewritten for better understanding of improvements by this study.

This point was highlighted as suggested in both the abstract and introduction.

It is also important to clearly mention that the BVOC emission module (i.e., process-based model) is not used in the EMAC ESM. For example, in introduction, the manuscript mentioned a few important

improvements in the LPJ-GUESS for BVOC simulations (e.g., process-based model for BVOC emission), but such process based model in LPJ-GUESS is not used in the EMAC. It makes us difficult to catch up the important works of this study.

This clarification has been made (L. 17, L. 53, L.59, etc.)

(3) Fig. 6 says that LPJ-GUESS produces no shrub in our earth which may be not true. I ask the authors to explain why there is no shrub land by LPJ-GUESS and implications for BVOC emission.

Shrubs are not included in the currently applied LPJ-GUESS global PFT set, consequently they are not considered in the applied simulation setup. Studies by Forrest et al. (2015) did not use explicit shrub PFTs as well, and only in more recent studies they are explicitly included (e.g. Allen et al., 2020). Even though this leads to less competition among some PFTs in certain regions, this is a limitation of the current study. However, including the new shrub PFTs is planned for future studies. This information has been added (L. 285)

In the manuscript we note that the lower magnitudes in monoterpene fluxes from MEGAN compared to ONEMIS result from the lack of representation of shrubs and needleleaf tree PFTs in LPJ-GUESS. These species are considered strong emitters of monoterpenes.

(4) This manuscript compares ONEMIS and MEGAN empirical BVOC models to LPJ-GUESS module in Fig. 9 and I ask the authors to include how to calculate BVOC emission in the LPJ-GUESS.

This information can be found in Section 2.2 "BVOC emission routine".

(5) The manuscript is saying that BVOC emission from ONEMIS and MEGAN is different (e.g., differences in tropical regions) but it will be better to explain (although that is simple) what makes such differences (e.g., LAI difference, temperature difference) and its implications for future climate simulations and BVOC emission uncertainties.

We state that isoprene emissions from MEGAN are higher in tropical regions, compared to ONEMIS emissions. These changes result from different canopy processes employed in ONEMIS and MEGAN respectively. ONEMIS and MEGAN emission values (e.g. Fig. 8, panel a and b) are coming from the same simulation, meaning that the input parameters in both modules (e.g. temperature, LAI, etc.) are identical. This is now clarified (L. 302).

(6) Such as Figure C1-C3 in Forrest et al. (2020), it will be better to quantify EMAC climate biases after improving the coupling processes between EMAC and LPJ-GUESS.

In this study we use the same climate variable coupling as in Forrest et al. (2020). Our coupling only involved vegetation information going into ONEMIS and MEGAN in EMAC. This means that the climate

biases are comparable to the ones presented in Forrest at al. (2020) and this information is not repeated in this study, but we refer to the previous analysis of the climate biases (L. 442).

(7) We feel that this study needs to mention previous studies to deal with dynamic vegetation model incorporated ESM (e.g., Levis et al., 2003) for clear understanding of what this study did. Also, it will be better if your work is compared to previous simulations (even though with simple or descriptive).

In the Introduction we mentioned various studies that incorporated vegetation representations in modelling BVOC emissions (including Levis et al. (2003)). We added a paragraph (L. 369) to compare our findings with results from Levis et al. (2003)

(8) Also, vegetation state such as LAI can be evaluated easily with remote sensed data such as MODIS and I am not sure if Fig. 4(d) corresponds to this reference data or not. Please provide more information on the data for Fig. 4(d) and please provide remote sensed LAI data for the reference.

The LAI product used in Fig. 4(d) (now Fig. 5) is indeed based on reference remotely sensed data (MODIS and AVHRR). This information is now included (L. 253).

**From the fourth referee response:**

The revision of the manuscript has been made suitable for publication through the major revision. This study developed a geoscientific model that calculates global BVOC emissions by connecting ONEMIS and MEGAN in EMAC modules to LPJ-GUESS, and tested the sensitivity of emissions through CO2 doubling experiments. I think that the experimental results of this study alone are scientifically meaningful findings and numbers. Here are some suggestions for minor fixes.

We thank the referee for the positive feedback and for pointing out mistakes in the text. Below we address the suggestions and minor fixes.

1. Please consider rephrasing sentences from Forrest et al., 2020, GMD - L90 91 and L105 108
- If there are more sentences, ...

We rephrased some text in the MS. Section 2.2 is based on an official LPJ-GUESS template - that's why some text is identical to Forrest et al. (2020). We now included a footnote referring to the official template.

2. Table 1 is hard to read. How about arranging it in one or two sentences?

A previous referee asked to provide more information about the difference between ONEMIS and MEGAN using a table. We however agree that the same information could be included in a paragraph as suggested.

3. Other minor comments are below:
@ abstract
- emissions from terrestrial vegetation, which represents
> emissions from terrestrial vegetation, which represent
- Please consider rephrasing this sentence:
and atmospheric chemistry is a recommended tool to address the fate of
> and atmospheric chemistry is recommended to address the fate of
- were found to be > were (delete "found to be")
- conclude that the proposed model setup is a useful tool for > conclude that the proposed model setup is useful for @L35
- the main precursor > the primary precursor

The MS was updated with all above-mentioned corrections.

**References**

Allen, J. R., Forrest, M., Hickler, T., Singarayer, J. S., Valdes, P. J., and Huntley, B.: Global vegetation patterns of the past 140,000 years, Journal of Biogeography, 47, 2073–2090, 2020.

Forrest, M., Eronen, J., Utescher, T., Knorr, G., Stepanek, C., Lohmann, G., and Hickler, T.: Climate-vegetation modelling and fossil plant data suggest low atmospheric $CO_2$ in the late Miocene, Climate of the Past, 11, 1701–1732, 2015.

Forrest, M., Tost, H., Lelieveld, J., and Hickler, T.: Including vegetation dynamics in an atmospheric chemistry-enabled general circulation model: linking LPJ-GUESS (v4. 0) with the EMAC modelling system (v2. 53), Geoscientific Model Development, 13, 1285–1309, 2020.

Levis, S., Wiedinmyer, C., Bonan, G. B., and Guenther, A.: Simulating biogenic volatile organic compound emissions in the Community Climate System Model, Journal of Geophysical Research: Atmospheres, 108, 2003.

---

## Author Response (AR4)

Author's response to editor comment (third round):

**"Isoprene and monoterpene simulations using the chemistry-climate model EMAC (v2.55) with interactive vegetation from LPJ-GUESS (v4.0)"**

5  by Ryan Vella et al.

We thank all editors for checking the revised manuscript and for the feedback provided. Here, the editor's comment (from December 23, 2022) is reproduced in black, while our comments are presented in blue.

**From the editor's response:**

10  Your revised manuscript is better to read now but our previous comments are fully satisfied. Please revise your manuscript several things more to improve readability.

1. Fig.1 helps what is done in this study and we more request that this manuscript give proper credit to Forrest et al. (2020). It is quite useful to add a sentence to describe that this study extends Forrest et al. (2020) by adding LAD for the BVOC emission simulations.

15  Manuscript was updated accordingly. Now it reads "This study focus on BVOC model processes in EMAC based on interactive vegetation from LPJ-GUESS. It extends on the model coupling between EMAC and LPJ-GUESS in Forrest et al. (2020) by employing new parameterisations to calculate the foliar density and leaf area density distribution from vegetation states in LPJ-GUESS." (L. 142)

2. To avoid misleading, please also add a sentence to clearly describe that the process-based model in
20  the LPJ-GUESS is not coupled into the ESM

This is now mentioned several times:

Fig. 1 and 2 captions.

L. 127: "BVOC emissions from this module are only calculated within the LPJ-GUESS model part and are not integrated into or transferred to EMAC at the current stage."

25  Now it reads (L. 127): "**Semi-process-based** BVOC emissions from **the LPJ-GUESS module** are only calculated within the LPJ-GUESS model part and are not integrated into or transferred to EMAC at the current stage."

L. 147: "The semi-process-based BVOC emissions from LPJ-GUESS are not integrated into EMAC and are only evaluated against the new vegetation-sensitive empirical-based emissions fluxes in EMAC."

30

3. It is confusing to mention the process-based model in the LPJ-GUESS in the manuscript (especially in introduction) because this study uses empirical models of ONEMIS and MEGAN.

Agreed. Process-based model descriptions from the introduction were removed. They are now briefly mentioned in Section 3.2 where semi-processed-based emissions from LPJ-GUESS are compared to
35  empirical-based emissions from ONEMIS and MEGAN in EMAC with interactive vegetation.

Also, I am not sure if the intercomparison of ONEMIS and MEGAN with the offline LPJ-GUESS provides differences between them only and not-quite-useful statements to mention other works.

The section comparing ONEMIS and MEGAN emissions with LPJ-GUESS emissions was amended with such considerations.

40  3. LAD, LAI and vegetation cover fraction show seasonal variations but figure captions do not provide any information on their timing. Also figure captions are not self-describing which makes us difficult to read.

The temporal (monthly) variability for LAI is show in Fig.5 panel (e) and discussed in L. 248. The monthly variabilities of the LAD and fractional converge are now discussed in L. 270 and L. 276, respectively.

45

All figure captions were revised making them self-describing.

4. It will be also good if this study mentions previous intercomparison studies between empirical and process-based models and their implications in this study.

This is now discussed in Section 3.2 "BVOC emissions from LPJ-GUESS" (L. 362-378).

**Further modifications:**

Fig. 2 was updated and now specifies that emissions in EMAC are only empirical-based and process-based emission from LPJ-GUESS are only used for comparison. The new vegetation variables derived in this study are now shown in red.

The Levis et al. (2003) study was moved from Section 3.3 to the Conclusions section.

Besides these comments, some text in the manuscript was modified to improve readability.

---

## Author Response (AR5)

Author's response to editor comment (technical corrections):

**"Isoprene and monoterpene simulations using the chemistry-climate model EMAC (v2.55) with interactive vegetation from LPJ-GUESS (v4.0)"**

by Ryan Vella et al.

We thank the topical editors for checking again the revised manuscript and for pointing out some technical corrections. Here, the editor's comment (from January 14, 2023) is reproduced in black, while our comments are presented in blue.

**From the editor's response:**

The code link provided in the manuscript is not available (10.5281/zenodo.6772205) and so please update the link following our journal guideline (https://www.geoscientific-model-development.net/about/manuscript_types.html#item1) Also please check if the coupled model code described in this paper is or will be published in the official MESSy update.

DOI link was updated to the "direct link" following the journal's guideline: https://doi.org/10.5281/zenodo.6772205. As explained in the *code availability* section, the model code remains restricted and can be only made available by the approval of the authors.

The code described in this work is indeed part of the current MESSy version. The following text was added: "The code described in this manuscript has already been incorporated into the official development branch of the EMAC modelling system and will therefore be part of all future released versions."